# Clearance of maternal barriers by paternal miR159 to initiate endosperm nuclear division in *Arabidopsis*

Youshang Zhao[1], Songyun Wang[1], Wenye Wu[1], Lei Li[1], Ting Jiang[1] & Binglian Zheng [1]

Sperm entry triggers central cell division during seed development, but what factors besides the genome are inherited from sperm, and the mechanism by which paternal factors regulate early division events, are not understood. Here we show that sperm-transmitted miR159 promotes endosperm nuclear division by repressing central cell-transmitted miR159 targets. Disruption of paternal miR159 causes approximately half of the seeds to abort as a result of defective endosperm nuclear divisions. In wild-type plants, *MYB33* and *MYB65*, two miR159 targets, are highly expressed in the central cell before fertilization, but both are rapidly abolished after fertilization. In contrast, loss of paternal miR159 leads to retention of MYB33 and MYB65 in the central cell after fertilization. Furthermore, ectopic expression of a miR159-resistant version of *MYB33* (mMYB33) in the endosperm significantly inhibits initiation of endosperm nuclear division. Collectively, these results show that paternal miR159 inhibits its maternal targets to promote endosperm nuclear division, thus uncovering a previously unknown paternal effect on seed development.

[1] State Key Laboratory of Genetic Engineering, Ministry of Education Key Laboratory of Biodiversity Sciences and Ecological Engineering, Institute of Plant Biology, School of Life Sciences, Fudan University, Shanghai 200438, China. These authors equally contributed: Youshang Zhao, Songyun Wang. Correspondence and requests for materials should be addressed to B.Z. (email: zhengbl@fudan.edu.cn)

Fertilization triggers the onset of development of a new generation, and the plant life cycle initiates from seed development, requiring the coordinated development of three structurally and genetically distinct compartments: the maternal seed coat and the concomitant development of two zygotic compartments, the embryo and the endosperm[1]. The endosperm plays critical roles in seed development by nourishing the developing embryo[1]. Endosperm development initiates rapidly after the central cell fertilization and undergoes multiple rounds of endosperm nucleus divisions without cell wall formation, resulting in a characteristic coenocytes-stage endosperm[1,2].

Many studies show that PcG (Polycomb Group) proteins negatively control endosperm proliferation independent of fertilization[3–8]. But it has been long enigmatic whether there are positive regulators triggering fertilization-dependent endosperm development. A previous study showed that sperm entry is sufficient to trigger endosperm nuclear division[9], indicating that the paternal genome and/or paternal factors are necessary for the endosperm development. A recent study uncovered that increasing auxin levels couples endosperm nuclear divisions to fertilization, in which paternal but not maternal TAR1 and YUC10, two key genes in auxin biosynthesis, were rapidly activated after fertilization[10]. These results demonstrate indeed the importance of specific paternal allele in the initiation of endosperm development. However, the first round of endosperm nuclear division occurs within several hours after fertilization[11], but paternal auxin biosynthetic genes were activated 10 h later after fertilization[10], indicating that parental factors other than locally synthesized auxin are involved in the initiation of endosperm nuclear divisions.

Due to the discrepancy in size between sperm and the egg cell, it is generally believed that sperm plays a very limited role in early development, besides providing their haploid genome. Therefore, many maternal factors have been demonstrated to be essential for embryo development[12,13]. However, it is increasingly clear that the male gamete carries additional coding or non-coding RNAs that are necessary for early development[14–16]. In mouse, interruption of sperm RNA delivery causes arrested zygotic divisions, resulting in embryonic lethality[17,18]. In contrast, there is only one example to show sperm-delivered SSP (Short Suspensor) RNA triggers the first zygotic division in plants[19].

miRNAs are 20–24 nucleotides small RNAs that play an essential role in various biological processes[20]. In animals, several studies show that sperm-delivered microRNAs (miRNAs) and/or small RNA-associated key components regulate early embryonic development[18,21,22]. Intriguingly, miRNA profiling analysis showed that many miRNAs are enriched in sperm relative to their levels in the vegetative cell of the pollen grain[23,24]. Among those sperm-enriched miRNAs, miR159 is the most remarkable[24]. The miR159 family is a very abundant miRNA family and represents one of the most ancient miRNAs in the plant kingdom[25], which has three members, miR159a, miR159b, and miR159c in Arabidopsis[26]. Bioinformatic and molecular analysis show that this family has eight target genes encoding R2R3 MYB domain proteins that are strongly conserved in both monocot and dicot species[27]. In general, miRNA and target genes have reciprocal expression pattern[20]. Unexpectedly, DUO1, one of miR159 targets, specifically accumulates in sperm[28,29], indicating that sperm-enriched miR159 is not destined to silence its target locally. Given that endosperm nuclear division initiates much earlier than zygotic division[9], we hypothesized that endosperm nuclear divisions might be preferentially dependent on factors inherited from parents. However, whether and how the paternal factors regulate early seed development, especially by which the mechanism endosperm nuclear division is initiated, is less well understood.

Given the enrichment of miR159 in sperm[23,24] and the decreased fertility caused by compromised miR159 activity[30,31], we used miR159 as an example to explore the function of paternally transmitted miRNA in seed development. Although both maternal and paternal miR159 are required for seed development, we show that paternal but not maternal miR159 is necessary for the initiation of endosperm nuclear divisions. Furthermore, we observed that the arrest of endosperm nuclear divisions was concomitant with the retention of MYB33 and MYB65, two major targets of miR159, in the central cell after fertilization, and we conclusively show that ectopic expression of miR159-resistant version of MYB33 in the central cell can phenocopy the arrested endosperm nuclear divisions seen in the absence of paternal miR159. These data uncover that a paternal miRNA initiates an early division event by removing the roadblock constituted by maternal factors.

## Results

**Paternal miR159 is required for seed development**. To investigate the biological function of sperm-enriched miR159, we took advantage of the mir159abc triple mutant, in which three independent T-DNAs were inserted in the pri-miRNA regions of MIR159a, MIR159b, and MIR159c, respectively, causing abolished miR159 accumulation[30,32]. A previous study showed short siliques with small and irregularly shaped seeds in the mir159abc mutant[30,32], we thus investigated whether and to what extent sperm-enriched miR159 plays a role in seed development.

We performed reciprocal crosses between Col-0 and mir159abc to define the effect of parental loss of miR159 on seed development. Based on the extent of reduced seed set, we characterized phenotype of seed development seen in F1 siliques of crosses into four grades: + represents silique with >90% seed set; ++ represents silique with 60–90% seed set; +++ represents silique with 30–60% seed set; ++++ represents silique with less than 30% seed set. Compared to that of hand-pollinated Col-0, ~30% of F1 siliques from ♀Col-0 × ♂mir159abc showed severely reduced seed set (Fig. 1a, b and Supplementary Table 1), and there was also ~28% of F1 siliques with moderately reduced seed set (Fig. 1a, b and Supplementary Table 1), indicating paternal miR159 is required for seed development. Moreover, when mir159abc was used as the female, Col-0 pollen partially rescued the phenotype of reduced seed set seen in self-fertilized mir159abc (Fig. 1b), further indicating the requirement of paternal miR159 in seed development. In contrast, although overall defects of seed development were comparable between mir159abc as the male and mir159abc as the female (Fig. 1b), only 10% of F1 siliques from ♀mir159abc × ♂Col-0 showed less than 30% seed set (Fig. 1b), implying that loss of maternal miR159 caused less severe effects on seed set than that of paternal miR159.

To further determine the deficiency of gametic transmission in the mir159abc mutant, we performed reciprocal crosses with Col-0 and mir159abc/+. We observed that both maternal and paternal transmissions were affected in mir159abc (Supplementary Table 2). Moreover, the efficiency of paternal transmission was comparable to that of maternal transmission (Supplementary Table 2). This reduced transmission is consistent with a reduced seed set defect and suggests that seed development requires both maternal and paternal miR159. Moreover, we show that compared to that of ♀mir159abc × ♂mir159abc, the seed size of ♀mir159abc × ♂Col-0 was slightly bigger (Fig. 1c and Supplementary Fig. 1a). Conversely, compare to that of ♀Col-0 × ♂Col-0, the seed size of ♀Col-0 × ♂mir159abc was slight smaller (Fig. 1c and Supplementary Fig. 1a), indicating that paternal miR159 is required for the control of seed size. Taken together, these results indicate that paternal miR159 is unexpectedly involved in seed

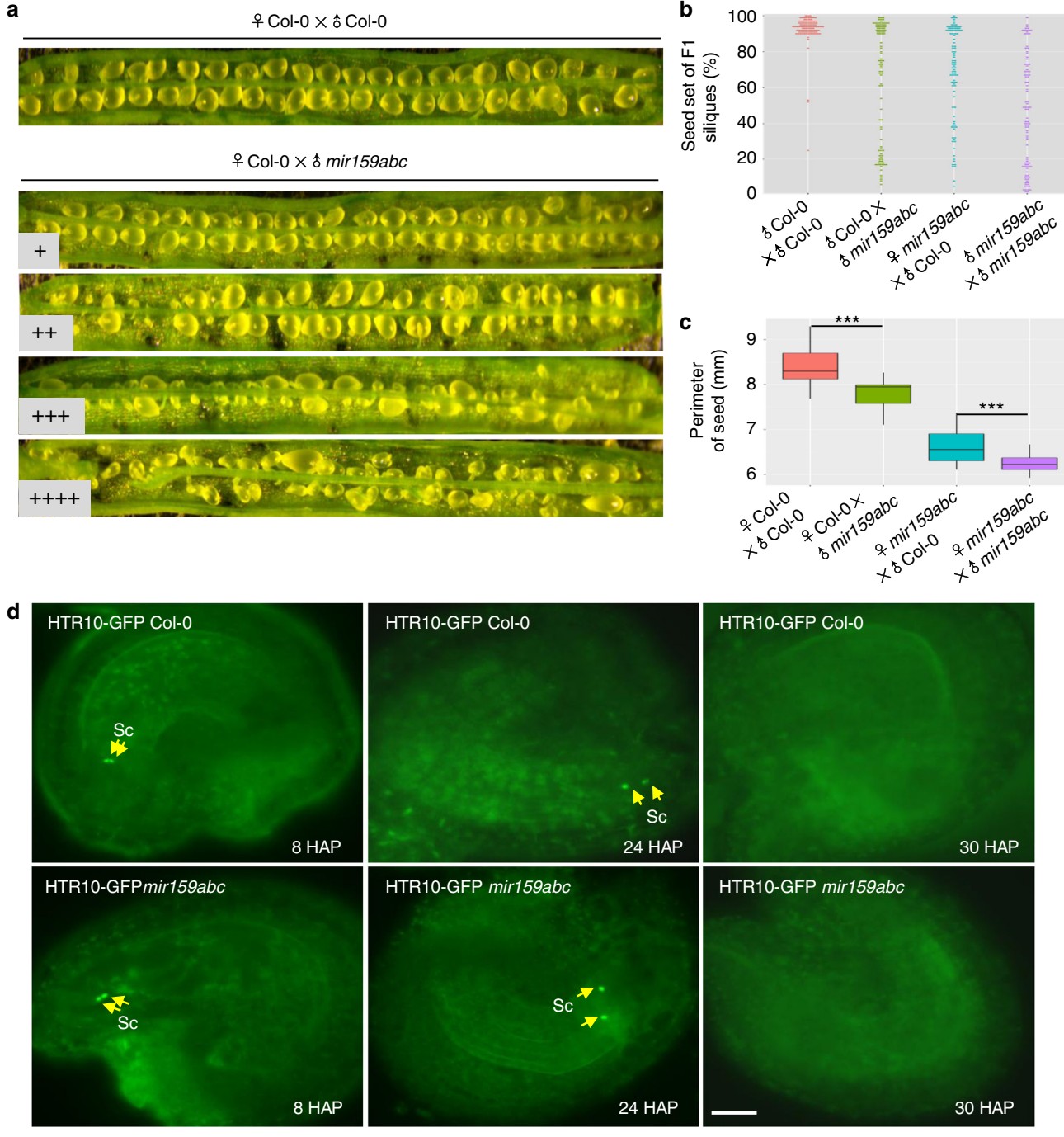

**Fig. 1** Both maternal and paternal miR159 are required for seed development. **a** Representative F1 siliques from 6 DAP (Day After Pollination) ♀Col-0 × ♂Col-0 and ♀Col-0 × ♂*mir159abc*, respectively. Smaller one from three lower panels indicates aborted seeds when Col-0 was pollinated by pollen of the *mir159abc* triple mutant. F1 siliques were divided into 4 categories according to % of seed set per silique: >90% (+), 60–90% (++), 30–60% (+++), <30% (++++), respectively. Source data are provided in the accompanying Source Data file. **b** Statistical analysis of seed set from **a**. The percentage of seed set per silique for each genotype is shown. One hundred siliques for each genotype were shown, and each dot represents one silique. **c** Statistical analysis of average perimeter per seed from each category. Perimeters of 20 seeds randomly picked from indicated genotypes were measured using the Image J software. Box plot shows the distribution of each seed perimeter for each genotype, and asterisks indicate a significant difference between the indicated samples (*t*-tests, *P*-value <0.05). **d** The release and migration of twin sperm in the developing seeds of Col-0 and the *mir159abc* triple mutant. Pistils from Col-0 and the *mir159abc* triple mutant harboring the HTR10-GFP transgene were self-pollinated, and the developing seeds at indicated time were dissected and imaged by fluorescence microscope. >50 developing seeds for each genotype were analyzed and representative images were shown. The yellow arrows indicate two released sperm. Embryo sacs are green due to auto-fluorescence. Scale bar, 20 μm

development. Considering that most studies have been focusing on the role of maternal effect on seed development, our following study will thus focus on the mechanism by which paternal miR159 is involved in seed development.

**Paternal miR159 triggers endosperm nuclear division**. To determine the step causing reduced seed set by disruption of paternal miR159, we first examined whether sperm cell formation, pollen tube growth and guidance, and sperm delivery and

migration were affected in the *mir159abc* mutant. DAPI staining showed that *mir159abc* pollen grain had normal twin sperm (Supplementary Fig. 1b). Decolorized aniline blue staining showed that pollen tubes of *mir159abc* had comparable pattern to that of Col-0 in growth rate, guidance, and burst (Supplementary Fig. 1c). The female gametophyte development of the *mir159abc* triple mutant had successfully formed the intact cell structure for double fertilization with two synergid cells, one egg cell, and one central cell, and three antipodal cells (Supplementary Fig. 1d). The release and migration of two sperm from the *mir159abc* triple mutant appear similar to that in wild-type plants (Fig. 1d). These results indicate that miR159 is not necessary for pollen development or sperm delivery before fertilization, and it is thus possible that the phenotype of reduced seed set caused by loss-of-paternal miR159 was due to abnormalities after fertilization.

Seed development is involved in the development of the maternal seed coat, the embryo, and the endosperm[1]. The initiation of endosperm nuclear division and the first zygotic division occur within 10 h and later than 30 h after pollination[1,11], implying that parental factors might be preferentially involved in endosperm nuclear divisions before the activation of zygotic genome. To determine whether paternal miR159 is required for early embryogenesis or endosperm nuclear divisions, we hand-pollinated Col-0 pistils with pollen of Col-0 or *mir159abc*, and then observed the process of endosperm nuclear divisions and embryo development using DIC (differential interference microscopy) imaging. At 12 HAP (hour after pollination), seeds from hand-pollinated Col-0 generally harbored 4 endosperm nuclei (Fig. 2a), indicating that endosperm had finished two rounds of nuclear divisions. In contrast, seeds from ♀Col-0 × ♂*mir159abc* showed great variation in endosperm nuclear divisions, in which 27.5% of endosperm nuclei arrested after fertilization (Fig. 2b), 24% of endosperm nuclei divided only once (Supplementary Fig. 2a), and only 48.5% of endosperm nuclei divided normally. Compared to 8 endosperm nuclei seen in self-pollinated Col-0 at 24 HAP (Fig. 2c), 25.3% of endosperm nuclei were clearly arrested in ♀Col-0 × ♂*mir159abc* seeds (Fig. 2d), and 27% of endosperm nuclei only divided twice (Supplementary Fig. 2b). Statistical analysis of number of endosperm nuclei at 24 HAP further confirmed the defects of arrested and/or delayed endosperm nuclear divisions in the loss of paternal miR159 (Fig. 2g). In contrast to more than 10 rounds divisions of endosperm nuclei in Col-0 at 48 HAP (Fig. 2e), endosperm nuclei in 23.7% of ♀Col-0 × ♂*mir159abc* seeds remained undivided (Fig. 2f), and 13.3% of endosperm nuclei only divided 2–3 rounds at 48 HAP (Supplementary Fig. 2c), further indicating that paternal miR159 is indispensable to promote endosperm nuclear divisions.

Because, the *mir159abc* triple mutant was obtained by genetic cross with three independent T-DNA insertion mutants, *mir159a*, *mir159b*, and *mir159c*[32], there is a possibility that multiple T-DNAs occasionally cause defective female development due to the T-DNA-mediated chromosome rearrangement[33]. If it is the case, the *mir159abc* triple mutant should theoretically cause much severe abnormalities than that of the double mutant. To test this point, we isolated the double mutant of *mir159ab*, *mir159ac*, and *mir159bc* from the progenies of the *mir159abc* triple mutant backcrossing with the wild-type Col-0 plants, respectively. Among three double mutants, the *mir159ab* plants exhibit indistinguishable morphology with that of the *mir159abc* triple mutant (Supplementary Fig. 2d), whereas other two double mutants (*mir159ac* and *mir159bc*) and all single mutants are indistinguishable from wild-type plants (Supplementary Fig. 2d), which was previously reported[32]. We then performed DIC analysis to investigate whether endosperm nuclear divisions in the *mir159ab* double mutant are affected as does in the *mir159abc*

triple mutant. Indeed, we observed that when the *mir159ab* double mutant was used as the pollen to pollinate Col-0 pistil, endosperm nuclear divisions of progenies were clearly arrested and/or delayed (Supplementary Fig. 2e), and the severity of defective endosperm nuclear divisions was comparable between the *mir159ab* double mutant and the *mir159abc* triple mutant (Fig. 2g). Therefore, we conclude that the defects of the *mir159abc* triple mutant as the pollen result in the compromised paternal miR159 activity, instead of side effects of multiple T-DNAs in the triple mutant.

To conclusively demonstrate that miR159 inherited from sperm is specifically required for initiation of endosperm nuclear divisions, we also compared the expression of proFWA::RFP, an endosperm marker[34], when either Col-0 or *mir159abc* was the male. As shown in Fig. 2h, proFWA::RFP was clearly expressed in the central cell before fertilization (the top panels). After fertilization, the RFP signals were progressively distributed in the dividing endosperm nuclei when Col-0 was the pollen donor (Fig. 2h, the middle panels). In contrast, the number of endosperm nuclei with RFP signals from the same developmental stage was significantly decreased when *mir159abc* was the pollen donor (Fig. 2h, the bottom panels). Compared to that all fused nuclei finished the first division when Col-0 was the male (Fig. 2h, the most left middle panel), almost all had not yet initiated endosperm nuclear division when *mir159abc* was the male (Fig. 2h, the most left bottom panel). At 12 HAP, there were still 60% of endosperm nuclei from *mir159abc* as the male remaining undivided status (Fig. 2h, the middle bottom panel). Moreover, even at 24 HAP, 31% of the central cell remains undivided (Fig. 2h, the right bottom panel). Collectively, we conclude that paternal miR159 plays a vital role in the initiation of endosperm nuclear divisions.

Notably, although maternal miR156 is required for early embryogenesis[35] and miRNAs might play an essential role in embryo development[36], the embryo development of seeds when *mir159abc* was the male was indistinguishable from that of Col-0 (Supplementary Fig. 2f,g), indicating that paternal miR159 had a minor role in embryogenesis. Taken together, these results indicate that paternally inherited miR159 was required for seed development by facilitating endosperm nuclear divisions.

**Paternal miR159 represses maternal targets in the seeds**. To understand the mechanism by which paternal miR159 promotes endosperm nuclear divisions, we hypothesized that sperm-transmitted miR159 represses its targets either in sperm locally or in the developing endosperm after fertilization. qRT-PCR analysis showed that relative to *MYB33*, *MYB65*, *MYB81*, and *MYB120*, *MYB97*, *MYB101*, *MYB120*, and *DUO1/MYB125* were significantly expressed in mature pollen (Supplementary Fig. 3a). In contrast, *MYB33* and *MYB65* were highly enriched in mature ovules (Supplementary Fig. 3b). Protein reporter fusions of MYB33 and MYB65 under the endogenous promoters confirmed these two miR159 targets are expressed in neither mature pollen nor sperm (Supplementary Fig. 3c), which is consistent with previous studies[37,38]. Unexpectedly, expression of all *MYBs* was not upregulated in mature pollen or mature ovules of the *mir159abc* mutant (Supplementary Fig. 3a,b), indicating that miR159-mediated target repression might be limited in gametes. Moreover, the fact that *DUO1* is specifically expressed in sperm[28] indicates that miR159 is not destined to repress its targets locally.

As miR159 has no role on the expression of target genes in sperm, and paternal miR159 is necessary for endosperm nuclear divisions, we wondered if sperm-transmitted miR159 represses target genes after fertilization? To test this hypothesis, we monitored the subcellular localization of MYB33 and MYB65

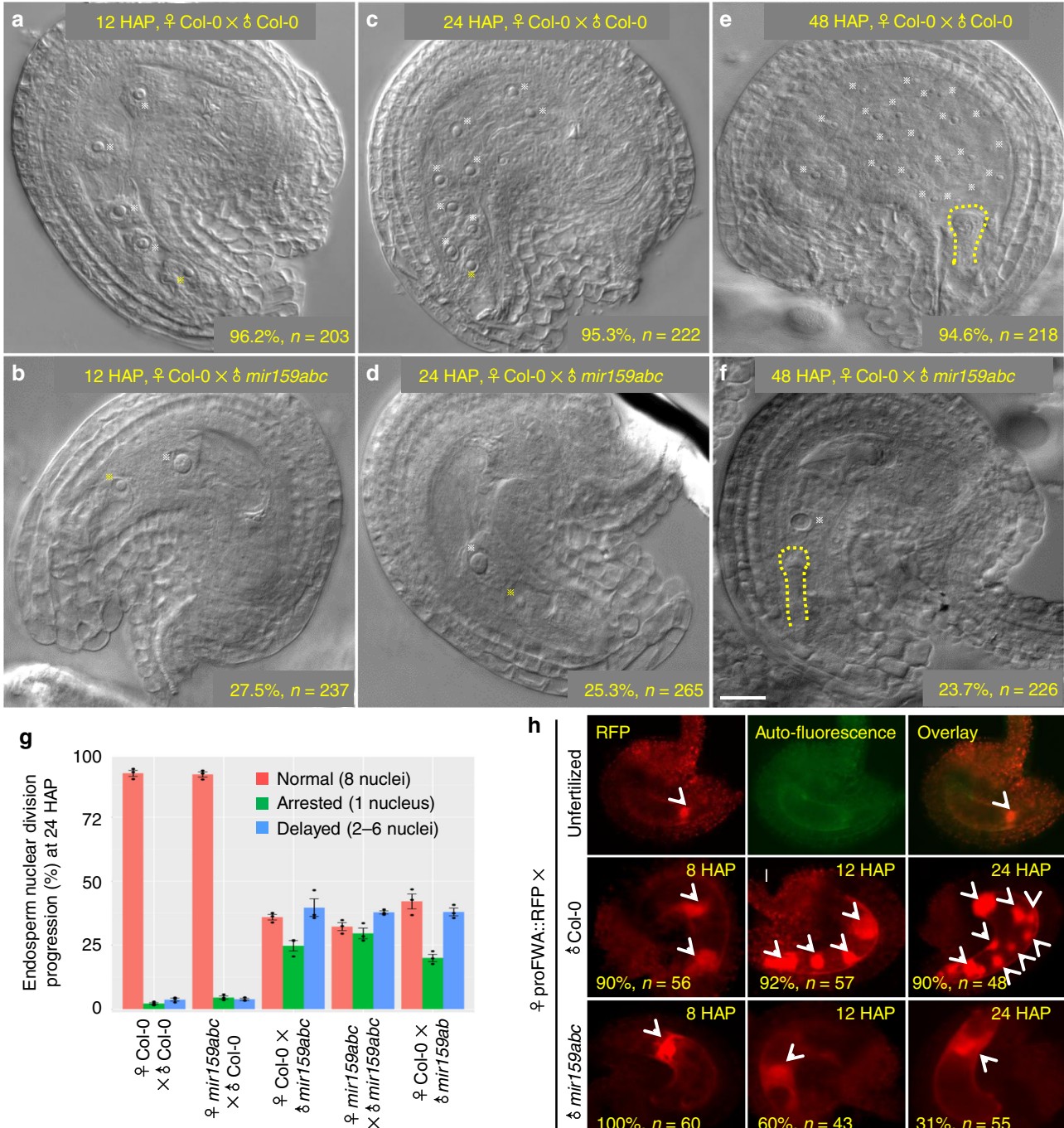

**Fig. 2** Lack of paternal miR159 caused defective endosperm nuclear divisions. **a–f** Endosperm nuclear divisions of the developing seeds from hand-pollinated Col-0 (**a**, **c**, **e**) or the F1 progenies of ♀Col-0 × ♂mir159abc (**b**, **d**, **f**) at indicated time (HAP, hour after pollination) by DIC imaging. White stars indicate endosperm nuclei or the central cell nucleus. Yellow stars indicate the fused nucleus by the egg cell and the sperm. Yellow dashed lines indicate the developing embryo. **g** Statistical analysis of endosperm nuclei number at 24 HAP. Seeds from hand-pollinated ♀Col-0 × ♂Col-0, ♀mir159abc × ♂Col-0, ♀Col-0 × ♂mir159abc, ♀mir159abc × ♂mir159abc, and ♀Col-0 × ♂mir159ab, respectively, were observed by DIC imaging and >200 F1 seeds from three biological replicates were examined for each category. Biological replicates for each genotype were shown by dots. **h** proFWA::RFP reporter in the unfertilized ovule (the top panels), as the female pollinated with Col-0 (the middle panels) or mir159abc (the bottom panels) at indicated time (HAP) by fluorescence microscopy. White arrows indicate RFP signals. Auto-fluorescence of the embryo sac is shown in the top middle panel. % indicates the ratio of developing seeds similar to the representative image relative to total ones. n represents total numbers of analyzed developing seeds for each category. Scale bar, 20 µm

driven by their endogenous promoters in the developing seed, because *MYB33* and *MYB65* are highly enriched in the mature ovule (Supplementary Fig. 3b). Fluorescence microscopic analysis showed that both MYB33-GFP and MYB65-RFP specifically accumulated in the central cell (Fig. 3a and Supplementary

Fig. 3d). Co-localization between MYB33-GFP and proFWA::RFP further confirms that MYB33 is significantly localized in the central cell (Fig. 3a). However, in contrast to that proFWA::RFP is highly enriched in the endosperm nuclei in the developing endosperm, MYB33-GFP (Fig. 3b) and MYB65-RFP

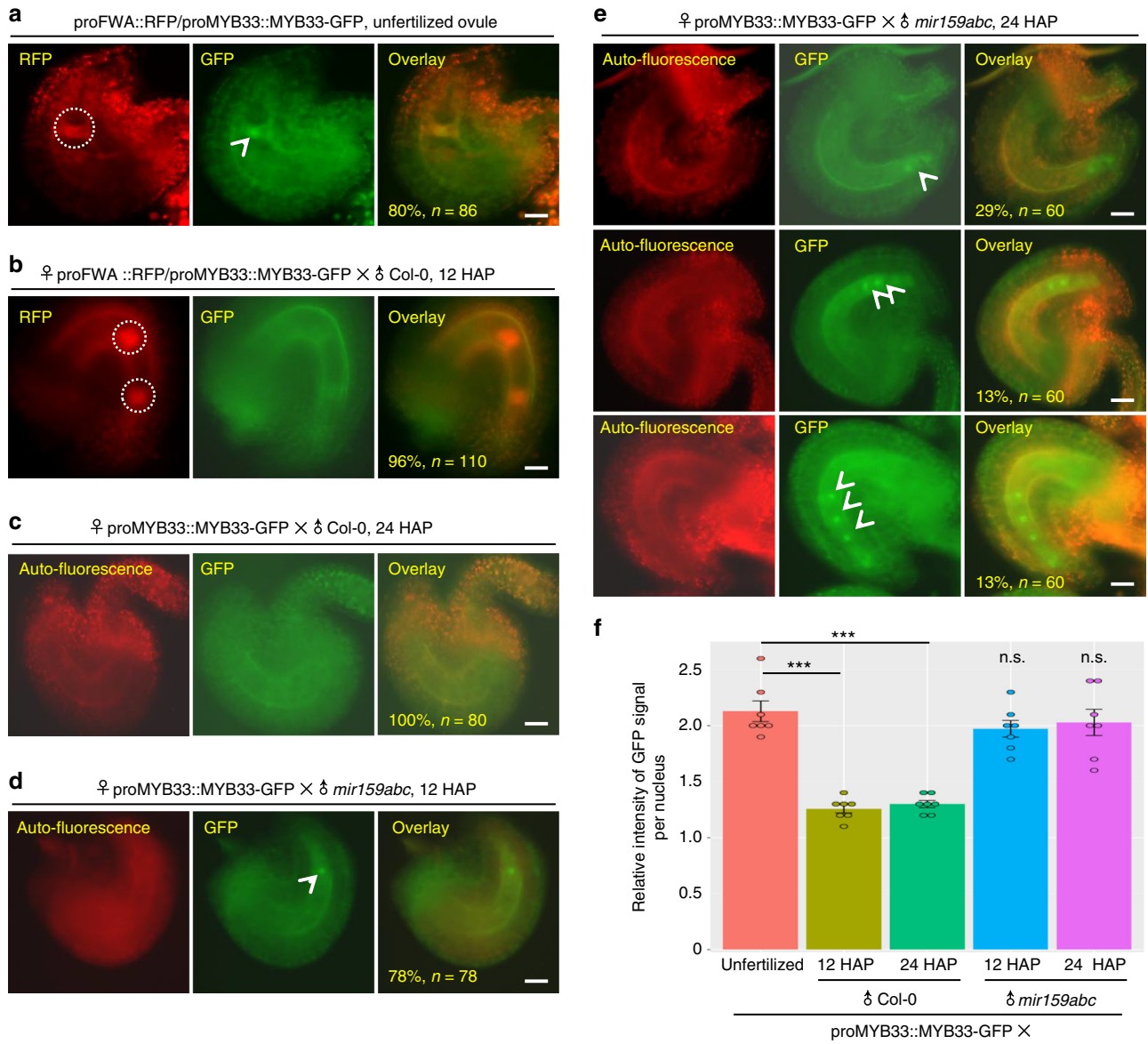

**Fig. 3** Subcellular localization of proMYB33::MYB33-GFP in the developing seeds. **a, b** Fluorescence images of the proMYB33::MYB33-GFP/proFWA::RFP doubly transgenic plants in the unfertilized ovule **a** and the developing seed at 12 HAP **b**. White dashed circles indicate the RFP signal, and the white arrowheads indicate the GFP signal. proFWA::RFP was used as the marker for the central cell/endosperm nuclei. Embryo sacs are green or red due to auto-fluorescence. **c–e** Fluorescence images of the developing seeds from the proMYB33::MYB33-GFP transgenic plants pollinated with pollen of Col-0 at 24 h **c**, mir159abc at 12 HAP **d** and 24 HAP **e**, respectively. All the left panels show the auto-fluorescence control of the developing seeds under the RFP channel, and the overlay was performed for each. White arrowheads indicate the GFP signal. % indicates the ratio of those developing seeds similar to the representative image relative to total ones. n represents total numbers of analyzed embryo sacs. Scale bar, 20 μm. **f** Quantification of proMYB33::MYB33-GFP signals from **a**–**e**. "unfertilized" indicates mature ovules of the proMYB33::MYB33-GFP transgenic plants as shown in the middle panel from **a**. Other columns represent that proMYB33::MYB33-GFP transgenic plants were pollinated as indicated time (12 and 24 HAP) by pollen of Col-0 or mir159abc, respectively. An arrow flanking each endosperm nucleus was used for the quantification of GFP signal intensity as shown in Supplementary Fig. 3d, and the measurements were performed using the Image-Pro Insight software. The GFP signal intensity was calculated by the number measured from the endosperm nucleus relative to the average number of flanking regions along the arrow. One small circle indicates the result from one image. Error bars show SD calculated from ~10 developing seeds for each category, and asterisks indicate a significant difference between the indicated samples (t-tests, P-value <0.05). n.s. indicates non-significant

(Supplementary Fig. 3d) were quickly attenuated after fertilization, as early as the first or second division of the central cell. As endosperm nuclear divisions proceeds, MYB33 becomes barely detected in the developing seed (Fig. 3c). In contrast, signals of MYB33-GFP (Fig. 3d, e) and MYB65-RFP (Supplementary Fig. 3d) in the undivided central cell and/or dividing endosperm nuclei are consistently visible after fertilization when mir159abc was the pollen donor. Quantitative analyses of fluorescence

signals of MYB33-GFP (Fig. 3f) or MYB65-RFP (Supplementary Fig. 3d) in the developing endosperm nuclei further confirm that the rapid removal of MYB33 and MYB65 after fertilization is controlled by paternal miR159.

**Paternal miR159 cleaves maternal *MYB33* in the seeds**. To conclusively demonstrate the rapid removal of MYB33 and MYB65 after fertilization is regulated by paternal miR159, we then

performed semi-quantitative 5′RACE RT-PCR to detect the 3′ cleavage products from MYB33 and MYB65 in the developing seeds. As shown in Fig. 4a, both MYB33 and MYB65 harbor miR159 binding sites with a predicted cleavage site in the 11th nucleotide of miR159 binding sites. Previous findings show that mutations of seed sequences (mutated MYB33, mMYB33) interrupt the cleavage of MYB33 and MYB65 by miR159[27,39]. If no miR159-mediated cleavage process occurs, PCR products using paired primers flanking the miR159 binding sites should be obtained (Fig. 4b). Otherwise, specific 3′ cleavage products should be obtained using the Generacer-specific primer and the primer from MYB33, or MYB65, or GFP sequences downstream of miR159 binding sites (Fig. 4b). We pollinated pistils of the wild-type plants Col-0 by using mature pollen from Col-0 or the mir159abc triple mutant, and collected the developing seeds for total RNA extractions. To detect the cleavage of mMYB33-GFP in vivo, we pollinated pistils of the mMYB33-GFP transgenic plants by using mature pollen from Col-0 plants. As expected, we show that PCR products flanking the miR159 binding sites from both MYB33 and MYB65 were obviously detected in the mature ovule before fertilization but not after fertilization (Fig. 4c), indicating that MYB33 and MYB65 are not cleaved in the ovule before fertilization.

Consistent with these results, no 3′ cleavage product was detected in the ovule before fertilization (Fig. 4c). In contrast, 3′ cleavage products of both MYB33 and MYB65 were significantly detected at 12 and 24 HAP (Fig. 4c). When the mir159abc triple mutant was used as the pollen, no 3′ cleavage products from both MYB33 and MYB65 were detected (Fig. 4c), and accompanied with significant accumulation of transcripts flanking miR159 binding sites of MYB33 and MYB65 in the developing seeds after fertilization (Fig. 4c). The mutation of seed sequences of miR159 binding sites of MYB33 (mMYB33-YFP) significantly abolished miR159-mediated cleavage process (Fig. 4d). Taken together, these results support that maternal MYB33 and MYB65 are repressed by paternal miR159 in the developing seeds.

The observation that only loss-of-paternal miR159 leads to retention of MYB33 and MYB65 in the central cell after fertilization indicates that neither maternal miR159 nor newly synthesized miR159 represses MYB33 and MYB65 after fertilization. To further exclude the possibility that MYB33 and MYB65 inherited from the central cell are repressed by newly synthesized miR159 after fertilization, we examined the promoter activities of three MIR159 genes in the developing seeds before and after fertilization. We introduced the proFWA::RFP into proMIR159::NLS-GFP

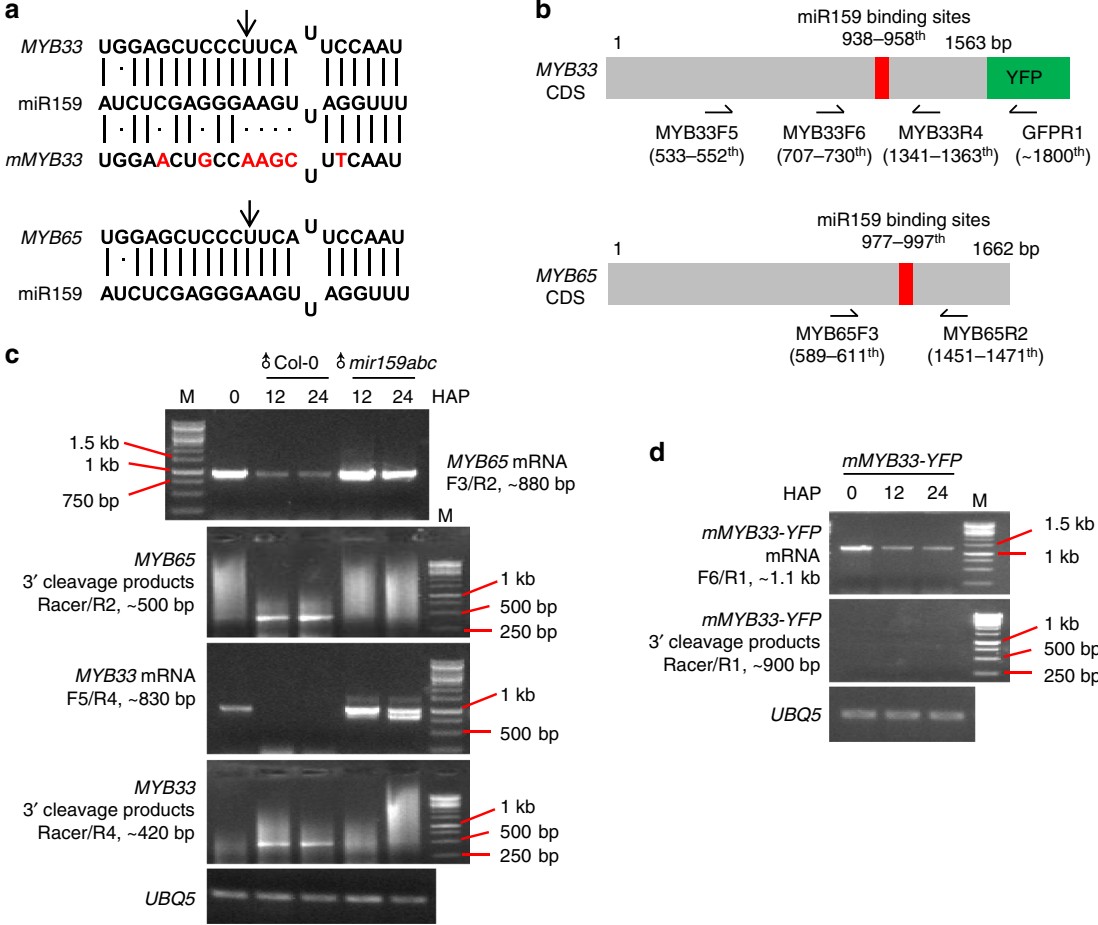

**Fig. 4** Detection of 3′ cleavage products of MYB33 and MYB65 by 5′RACE-PCR. **a** Alignment of miR159 and its targeting sites on MYB33 and MYB65. Mutated nucleotides of miR159 binding sites are highlighted in red as mMYB33. Vertical lines and dots indicate complementary nucleotides and uncomplimentary one, respectively. The arrows indicate the cleavage site of miR159 on the coding region of MYB33 or MYB65. **a** Schematics of MYB33 and MYB65 coding regions, the location of miR159 binding sites, positions of primers used in **c–d**. Red rectangles indicate miR159 targeting sites. Arrows indicate the orientation of primers. **c**, **d** Semi-quantitative 5′RACE-PCR to detect 3′ cleavage fragments generated through miR159-guided cleavage of MYB33 and MYB65 transcripts. UBQ5 is a loading control. M indicates the DNA ladder, and standard sizes were labeled. For **c**, we used pistils from Col-0, and pollinated it using pollen from Col-0 or the mir159abc triple mutant. For **d**, we used pistils from the proFWA::mMYB33-GFP transgenic plants, and pollinated it using pollen from Col-0. The experiments were performed with two biological replicates. Source data are provided in the accompanying Source Data file

transgenic plants by *Agrobacteria*-mediated transformation, in which proFWA::RFP was used as the central cell/endosperm marker. By comparing with the signal of proFWA::RFP, we observed that all *MIR159a* (Supplementary Fig. 4a), *MIR159b* (Supplementary Fig. 4b), and *MIR159c* (Supplementary Fig. 4c) were weakly expressed in the central cell before fertilization. However, in contrast to the strong activity of the *FWA* promoter in the dividing endosperm nuclei (Supplementary Fig. 4a-c), transcription of all three *MIR159* genes in the developing seeds was barely detectable after fertilization (Supplementary Fig. 4a-c), indicating that *MIR159* transcription is inactive during early seed development. Quantitative analyses of GFP signals in the developing endosperm nuclei (Supplementary Fig. 4d) further confirm that locally synthesis of miR159 in the developing seeds is limited. Taken together, these results demonstrate that paternal miR159 inhibits maternally inherited MYB33 and MYB65 in the developing seeds to promote endosperm nuclear divisions.

**Over-expression of *MYB33* blocks endosperm nuclear divisions.** To fully demonstrate that the rapid removal of MYB33 and MYB65 after fertilization is a prerequisite for initiation and progression of endosperm nuclear divisions, we hypothesized that over-expression of versions of *MYB33* or *MYB65* with mutations at the miR159 binding sites should theoretically mimic the phenotype of defective endosperm nuclear divisions caused by compromised activity of paternal miR159. To test this hypothesis, we made a construct of mMYB33::YFP, in which the central cell/endosperm-specific promoter of *FWA* drove expression of *MYB33* with the mutated miR159 target sites without changing amino acids, in frame with the YFP reporter gene (Fig. 5a), and generated 18 independent transgenic plants (mMYB33-OE). For comparison, transgenic plants were also generated with wild-type MYB33-YFP driven by the *FWA* promoter (MYB33-OE). Similar to that of pMYB33::MYB33-YFP transgenic plants, both MYB33-YFP and mMYB33-YFP were highly expressed in the central cell before fertilization (Supplementary Fig. 5a-c), and MYB33 but not mMYB33 was depleted after fertilization when Col-0 was the pollen donor (Supplementary Fig. 5a-c), indicating that the mutation of miR159 binding sites blocked the repression of *MYB33* by miR159. The retention of both MYB33-YFP and mMYB33-YFP in the central cell when the *mir159abc* mutant was the pollen donor (Supplementary Fig. 5a-c), further indicates that paternal miR159 is necessary for the repression of MYB33 in the central cell after fertilization.

To investigate whether retention of MYB33 in the central cell have consequences for endosperm nuclear divisions, we compared the initiation and progression of endosperm nuclear divisions between MYB33-OE and mMYB33-OE transgenic plants by DIC analysis. Compared to the normal progression of endosperm nuclear divisions in MYB33-OE transgenic plants (Fig. 5b, c), the initiation of endosperm nuclear division from 17 out of 18 individual mMYB33-OE transgenic plants was arrested (Fig. 5d–g). At 12 HAP, >97% of the central cell from the MYB33-OE plants finished two rounds of nuclear divisions (Fig. 5b). In contrast, there were approximately one-third of the developing seeds from the mMYB33-OE plants without endosperm nuclear division (Fig. 5d) or with only once endosperm nuclear division (Fig. 5e). Even though until 24 HAP, 29.9% of the developing seeds from the mMYB33-OE plants failed to initiate endosperm nuclear division (Fig. 5f). The defects of endosperm development of mMYB33-OE plants are similar to that of the *mir159abc* triple mutant, further supporting that retention of MYB33 in the central cell after fertilization is the barrier of the initiation of endosperm nuclear divisions.

Coincident with the essential role of endosperm nuclear divisions in seed development, we show that 17 out of 18 individual mMYB33-OE transgenic plants harbored aborted seeds (Fig. 5h), whereas all MYB33-OE transgenic plants exhibited fully developed seeds (Fig. 5h). The reduced seed set seen in self-pollinated mMYB33-OE transgenic plants was variable from line to line, but, on average, these plants had 20–45% reduced seed set (Supplementary Fig. 5d). Taken together, these results indicate that over-expression of a miR159-resistant *MYB33* in the central cell after fertilization is able to mimic the phenotype of defective endosperm nuclear divisions caused by loss-of-paternal miR159.

**Discussion**

Both in animals and plants, parental gene products delivered into the newly fused zygotic cell regulate embryonic development before activation of the zygotic genome[40]. Plant development is initiated from seed development, in which the endosperm nuclear division occurs earlier than zygotic division, implicating that parental factors might play much more important roles in early endosperm development. However, how the newly formed cell fused by the central cell and the sperm perceives these factors to initiate the first nuclear division is less well understood. We reveal that miR159 is the first identified molecule transmitted from sperm that is required for seed development by triggering endosperm nuclear divisions, in which we provide multiple lines of evidence supporting this conclusion: (i) not only maternal miR159, but also paternal miR159 is required for seed development, (ii) loss-of-paternal miR159 causes defective initiation of endosperm nuclear divisions, (iii) maternal MYB33 and MYB65 are quickly eliminated by paternal miR159 after fertilization, (iv) ectopic expression of MYB33 mimics arrested endosperm nuclear division in loss-of-paternal miR159. Collectively, our data strongly support the conclusion that miR159 transmitted from sperm triggers endosperm nuclear division by clearing the maternally inherited roadblock. Together with that mouse sperm-borne miRNAs are required for zygotic division[18,22,41], these findings imply that the involvement of sperm-delivered miRNA in early embryogenesis might be evolutionarily conserved.

Previous studies showed that miRNA activity is globally suppressed in mouse oocytes[42,43]. The observations that the co-existence of *DUO1*/miR159 in sperm[23] and no upregulation of *MYBs* in both pollen and the developing seeds of the *mir159abc* mutant (Supplementary Fig. 3a,b), indicate that limited miRNA activity in gametes is also highly conserved from plants to animals. The fact that AGO1, the key regulator for miRNA-mediated target repression, is very lowly expressed in gametic cells[23,37,44], might explain the observations of limited miRNA activity in plant gametes. However, miR159 is able to inhibit MYB33 and MYB65 in the developing seeds after fertilization (Fig. 3 and Fig. 4), indicating that AGO1 or other AGOs redundant with AGO1 might be rapidly activated after fertilization.

The remarkable role of paternal miR159 in endosperm development, but not in embryogenesis could be explained because the first zygotic division is much later than the first division of the endosperm nucleus[40,45]. Although paternal miR159 has undetectable role in embryo development in our case, it is worthy to investigate whether maternal miR159 is involved in early embryo development, as disrupted transmission of maternal miR159 also caused defective seed development (Fig. 1b, c). Notably, multiple T-DNAs-mediated chromosome translocation occasionally occurs and finally causes defective female gametophyte development[33]. To test whether this is the case for the *mir159abc* triple mutant, we observed the ovule morphology and found that the mature ovule of the *mir159abc* triple mutant harbors normal seven-cell structure (Supplementary Fig. 1d). Moreover, DIC analysis of the *mir159ab* double mutant show defective endosperm nuclear divisions similar to that of the *mir159abc* triple

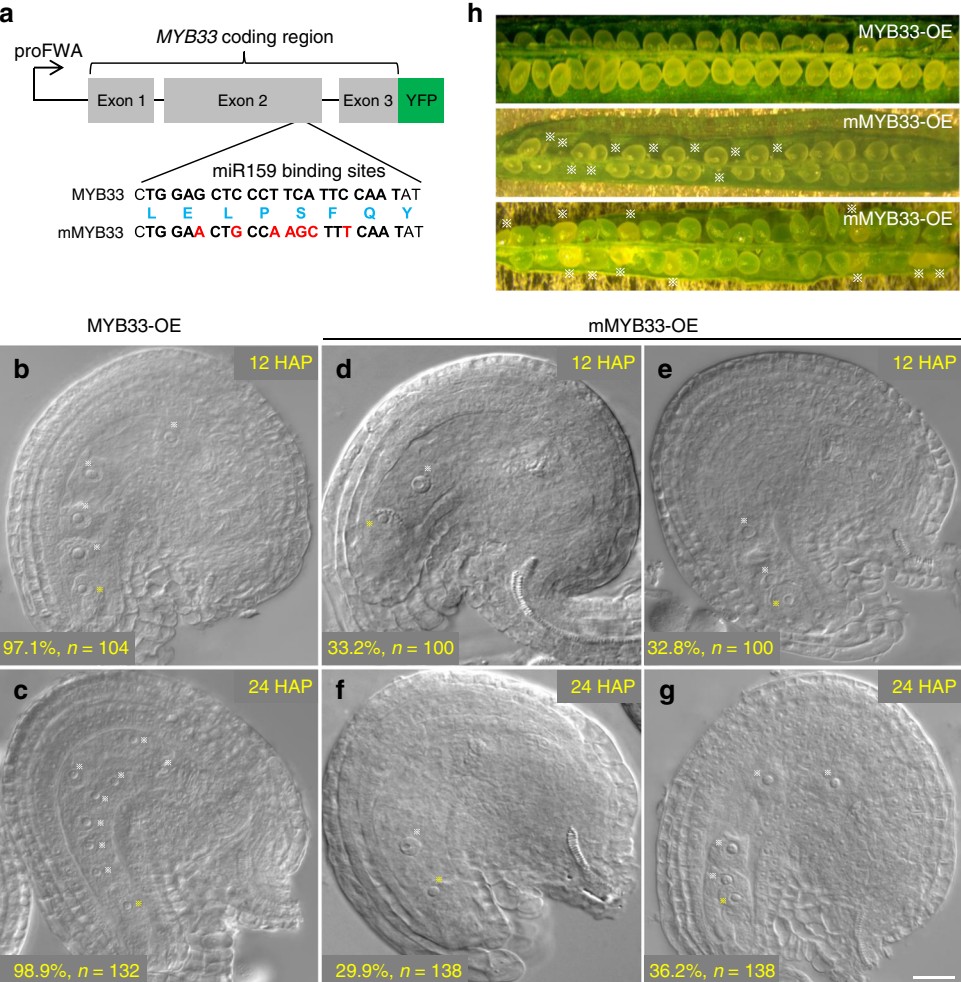

**Fig. 5** Over-expression of *mMYB33* causes defective endosperm nuclear divisions. The proFWA::MYB33-YFP (MYB33-OE) or proFWA::mMYB33-YFP (mMYB33-OE) transgene driven by the FWA promoter made in the vector pB7YWG2. The alteration to the nucleotide sequence in the miR159 binding sites of *MYB33* is shown, and nucleotides highlighted in red for *mMYB33* are those differing from the wild type (*MYB33*). The unaltered amino acid sequences are shown in blue. The nucleotides in bold indicate the 21 nt miR159 binding sites. Gray boxes indicate three exons of *MYB33*. **b–g** DIC imaging of endosperm nuclear divisions from MYB33-OE **b**, **c** and mMYB33-OE **d–g** transgenic plants at indicated time (HAP), respectively. White stars indicate endosperm nuclei or the central cell nucleus. Yellow stars indicate the fused nucleus by the egg cell and the sperm. % indicates the ratio of developing seeds similar to the representative image relative to total ones. *n* represents total numbers of observed developing seeds for each category. Scale bar, 20 μm. Representative images for dissected seeds from MYB33-OE or mMYB33-OE transgenic plants. Normal seeds after 6 days are oval and dark green, white stars indicate aborted seeds. Source data are provided in the accompanying Source Data file

mutant (Fig. 2g and Supplementary Fig. 2e), further indicating that the defective endosperm development in the *mir159abc* triple mutant is due to defective paternal miR159 activity. However, as three *MIR159* genes were obviously detected in the central cell before fertilization (Supplementary Fig. 4a-c) and loss-of-maternal miR159 caused seed abortion (Fig. 1), the investigation on the role of maternal miR159 will provide additional information on the biological significance of parentally transmitted miRNA in early seed development.

As our finding that paternal miR159-dependent repression of maternal MYB33 and MYB65 after fertilization is necessary to initiate endosperm nuclear division, the mechanism by which maternal MYB33 and MYB65 inhibit the central cell division triggered by fertilization remains unknown. A previous study showed that MYB33 and MYB65 negatively regulate cell proliferation in vegetative tissue[46], indicating the potential role of MYB33 and MYB65 as a molecular switch in dividing cells. In addition, loss of activity in the PcG proteins caused uncontrolled endosperm nuclear divisions independent of fertilization[47]. A

recent study showed that PcG proteins repress maternal auxin biosynthesis genes, *YUC10* and *TAR1*, thus allowing the expression of its paternal allelic genes to synthesize auxin, which is sufficient to promote endosperm nuclear divisions[10]. However, the first division of endosperm nucleus occurs around 10 h after pollination[11], but paternal auxin biosynthetic genes are activated around 24 h after pollination[10], indicating that factors other than newly synthesized auxin trigger early endosperm nuclear division. Our finding demonstrates that miR159, directly transmitted from sperm, is able to initiate endosperm nuclear division within 12 h after pollination. A recent study shows that maternally supplied auxin directly stimulate early embryo patterning[13], indicating that parentally transmitted factors other than the haploid genome play an essential and conserved role during early division events.

## Methods
**Plant material and growth conditions**. Col-0 was used as the wild-type plants. Transgenic plants of proMYB33::MYB33-GFP, proMYB65::MYB65-RFP, proFWA::MYB33-YFP, proFWA::mMYB33-YFP, and proFWA::RFP are the Col-0

background which were made by our lab. The reporter lines of *proMIR159a::GFP*, *proMIR159b::GFP*, and *proMIR159b::GFP* are referred[29]. The reporter line of HTR10-GFP (CS67829) was ordered from ABRC. The *mir159abc* triple mutant was kindly provided by Dr. Anthony A. Millar, which was constructed by three independent T-DNA insertional mutants (SAIL_430_F11 for *mir159a*; SAIL_770_GO5 for *mir159b*; and SAIL_248_G11 for *mir159c*)[32]. The mutant and wild-type seeds were grown in the growth room with humidity of 65% under a 16 h light/8 h dark cycle at 22 °C.

**Plasmid construction.** To generate the *proMYB33::MYB33-GFP* construct, the *MYB33* genomic region from 1832 bp upstream of the ATG to the end of ORF (without stop codon) was amplified from Col-0 genomic DNA with primers MYB33F1/MYB33R1, cloned into pENTR-D/TOPO, and then transferred into pMDC107. Similarly, the *MYB65* genomic region containing the coding sequence (without the stop codon) and 2449 bp upstream was amplified with primers MYB65F1/MYB65R1, cloned into pENTR-D/TOPO, and then transferred into pMDC163-RFP. To generate the *proFWA::RFP* constructs, the promoter sequences of *FWA* was amplified from Col-0 genomic DNA using primers FWAF1/R1, cloned into pENTR1A, and then transferred to the pMDC163, in which the GUS sequences was replaced by the RFP sequences. To generate the proFWA::MYB33-YFP construct, coding region of *MYB33* was amplified from Col-0 cDNA using primers MYB33F2/R2, cloned into pENTR1A, and then transferred to the pB7YWG2 in which the 35S promoter was replaced by the promoter of FWA, respectively. For proFWA::mMYB33-YFP, the seed sequences of miR159 binding sites in the CDS of *MYB33* were mutated using primers MYB33F3/R3 according to[39]. Primer information is listed in Supplementary Table 3.

**Genetic cross.** Due to the delay in vegetative growth of the *mir159abc* mutant, 8-week-old mutant plants were used for crosses, whereas Col-0 plants were 5 weeks old. Flowers at stage 12 were emasculated and pistils were left to grow for ~12 h for maturation. Then pistils were hand-pollinated with pollen grains of Col-0 or the *mir159abc* mutant. For reciprocal cross, Col-0 and *mir159abc/+* were crossed. For calculating the ratio of aborted seeds, pistils at 6 days after pollination (DAP) were dissected and the numbers of normal and aborted seeds were counted under Leica dissecting microscope.

**Microscopy.** For examination of embryo development with differential interference contrast (DIC) microscopy, the developing seeds >=2 days after pollination (DAP) were mounted in clearing solution (chloral hydrate:water:glycerol, w/v/v: 8:3:1) for DIC imaging with UPlanFLN ×20 objective. For examination of endosperm development, the developing seeds were fixed in FAA fixative solution (3.7% formaldehyde, 5% acetic acid, and 50% ethanol) for 6–8 h and then mounted in clearing solution for DIC imaging with UPlanFLN ×40 objective. The microscope used was an Olympus BX53 equipped with a Sony ICX285 CCD camera. For fluorescence analysis, the developing seeds were dissected and were immediately mounted in water. Fluorescence microscopy analysis was carried out with an Olympus BX53 microscope (image acquisition software: Image-Pro Insight software; objective: UPlanFLN ×40). DAPI staining of pollen was examined under UV channel. Decolorized aniline blue staining of in vivo pollen tubes was examined under UV channel using UPlanFLN ×4 objective. Images were further processed using Adobe Photoshop and Image J.

**RT-PCR and qRT-PCR analysis.** Open flowers and the developing seeds from dissected siliques at indicated time points were collected in nitrogen liquid immediately, and total RNA was extracted with Trizol reagent. Five micrograms of total RNA was treated with RQ1 DNase for 1 h at 37 °C and 1–2 μg Dnase I-digested RNA was reversely transcribed using PrimeScript II Reverse Transcriptase and Oligo d (T). cDNA was diluted 1:10 before the qPCR reaction. qRT-PCR assays were performed using iQ™ SYBR Green Supermix and Bio-Rad CFX96 Real-Time PCR detection system. Primer information is shown in Supplementary Table 3.

**5′RACE-PCR experiments.** For the detection of 3′ cleavage products from miR159-targeted *MYB33* and *MYB65*, 5′ RACE was performed using the Gen-eRacer kit (Invitrogen). Flowers at stage 12 from Col-0 were emasculated and naked pistils were left to grow for ~12 h for maturation. Then naked pistils were hand-pollinated with pollen grains of Col-0 or the *mir159abc* mutant. For detecting the cleavage products of mMYB33-GFP transcripts, the naked pistils from proFWA::mMYB33-GFP transgenic plants were hand-pollinated with pollen grains of Col-0. The developing seeds from dissected siliques indicated time points were collected in liquid nitrogen immediately, and total RNA was extracted with Trizol reagent. A total of 1–5 μg total RNA was directly ligated with the GeneRacer RNA oligo nucleotide and reverse transcription reactions were carried out with gene-specific primers listed in Supplementary Table 3. Semi-quantitative PCR reactions were performed to quantify 3′ cleavage products and full length transcripts, respectively. The PCR reactions were performed as regular conditions: 95 °C 30 s, 58 °C 30 s, 72 °C 1 min, 32 cycles. The same amount of RNA was reversely transcribed with oligo dT to amplify *UBQ5* as an internal loading control.

## Data availability

All data generated or analyzed during this study are included in this published article (and its Supplementary Information Files), and the data that support the findings of this study are available from the corresponding author upon reasonable request. Underlying data for Fig. 1b, c, Fig. 2g and Fig. 3f, as well as original silique and RACE-PCR gel images, are provided as a Source Data file. A reporting summary for this Article is available as a Supplementary Information file.

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

## Acknowledgements

We are grateful to Profs. Hong Ma, Xiuren Zhang, and Sheila McCormick for critical reading of the manuscript. We thank Anthony A. Millar for kindly providing the seeds of the *mir159abc* mutant. We appreciate Dr. Shengben Li for kindly providing some 5′ RACE-PCR reagents. We thank Xiaotuo Zhang for help in figure preparations. This work was supported by the National Natural Science Foundation of China (31830045, 31470281, 31671261) and the Recruitment Program of Global Expects (China).

## Author contributions

Y.Z. performed most of experiments. S.W. performed many preliminary experiments. W.W., L.L. and T.J. helped in genotyping and DIC imaging. Y.Z. and B.Z. designed the experiments and analyzed the data; S.W. and B.Z. conceived the project and wrote the article.

## Additional information

**Competing interests:** The authors declare no competing interests.

