## [Peer Review File · Nature Communications]

Reviewers' Comments:

Reviewer #1:

Remarks to the Author:

This is a review of the manuscript entitled "Clearance of maternal barriers by paternal miR159 to initiate endosperm nuclear divisions in Arabidopsis" submitted to Nature Communications by Wang et al. In this manuscript, the authors demonstrate that paternal miR159 is essential to promote endosperm nuclear divisions just after fertilization. Without the paternal miR159 contribution, endosperm divisions are delayed, leading to partial sterility. The authors also identify two key targets of paternal miR159, which are maternally-expressed MYB genes. They demonstrate that miR159-resistant MYB genes provide the same endosperm phenotype as a miR159-mutant paternal contribution. Overall, the paper aims to demonstrate a rapid clearing of a maternal transcript by paternal sperm microRNAs at the moment of fertilization.

Overall, I found this manuscript very interesting, but I found the key data was often buried or very difficult to see. I have split my comments into major issues to be addressed, and smaller changes to be made.

Major issues to be addressed

1. There are several figure panels that require quantification, rather than relying on qualitative visual inspection of the data.
 - A. The major weakness of this manuscript is Figure 3. I cannot tell what the authors want me to observe, and I cannot draw their same conclusions from the images. This may be a failure of communication rather than anything wrong with the data. Perhaps show the reader what is background auto-fluorescence vs what is real RFP/GFP. Or perhaps add more annotation to the figures, such as in Figure 2. Also, Quantification is needed.
 - B. I have the same comment for Supplemental Figure 3D and Figure S4. I do not agree with the conclusions simply because I am not seeing the same thing as the authors are seeing.
 - C. Figure 1C – quantification is required.
2. I would not continue to refer to miR159 as "essential" for seed development. Some seeds are produced from a miR159 mutant x mutant cross (which is how the authors generate their key homozygous mutant line), and therefore I would write that fertility is decreased, but it is not "essential".
3. In the discussion, please speculate on why miR159 does not act on DUO1. Why are microRNAs ineffective in the gametes, but then upon fertilization they now function? This point is raised several times in the manuscript, and why/how needs to be at least speculated on in the Discussion section.
4. I find the choice of expression of DDM1 in the endosperm as a fluorescent marker very odd. Please explain why this was performed rather than using H2B or a simple NLS tag to make YFP nuclear-localized?
5. Some of the key data should be moved from the Supplemental section to the main figures: Figure S4, Figure S2C.
6. For the mMYB33-YFP transgene, details were missing in the construction: Is the mMYB33 protein identical to the MYB33 protein? Their nucleic acid sequences must be different (with and without miR159 target sites), but are there other changes that could account for their expression differences after fertilization?
7. Please perform 5' RACE to detect MYB33 & MYB65 miR159 cleavage products in the developing seeds and pollen. The detection of microRNA cleavage (and lack thereof in the mMYB33 line) would greatly strengthen the authors' arguments.

Smaller changes to be made

1. The first sentence of the last paragraph of the Introduction needs to be rewritten with improved clarity.
2. Supplemental Figure 1C needs to be shown with a wild-type control or reference sample. Since I do not work on pollen, I do not have a frame of reference of what a normal pistil should look like,

so I cannot determine if this image is normal or not.

3. The first Results section is not critical for the rest of the manuscript, so I would suggest shortening it.

Reviewer #2:

Remarks to the Author:

Clearance of maternal barriers by paternal miR159 to initiate endosperm and nuclear divisions in Arabidopsis.

Songyun Wang et al.

In flowering plants double fertilization of the egg and central cell by two independent sperm cells triggers the initiation of zygotic embryogenesis and endosperm development. Although both embryo and endosperm formation have been shown to be under the influence of many maternal factors, paternally inherited factors playing a role in seed formation have remained elusive, and no specific paternally-derived microRNA (miRNA) capable of repressing the function of maternally derived transcripts has been identified. In this manuscript Wang et al. have investigated the function of the miR159 family of miRNAs during endosperm development. They provide evidence showing that reciprocal crosses to a triple miR159abc insertional mutant have defects in seed formation and endosperm nuclei proliferation when mir159abc is used as a pollen donor. They also show that the miR159 predicted targets MYB33 and MYB65 are expressed during female but not male gametogenesis prior to fertilization, and claim that miR159 activity is responsible for abolishing their activity following fertilization. A transgenic Arabidopsis line expressing a version of MYB33 that has a mutation in the MIR159 target site shows defects in free nuclear proliferation, on the basis of which the authors claim that miR159 triggers endosperm nuclear divisions through repressing the central cell-delivered miR159 targets activates endosperm proliferation through repressing its direct targets in the central cell.

I think this is a relevant topic and study that could be of wide interest for the scientific community working on early seed development. It undoubtedly presents new findings; however, in my opinion the work is not fully convincing in its present form. Here is a list of further experimental evidence and suggested modifications that could strengthen the conclusions:

1) The genetic analysis of the miR159 mutant(s) is limited to the triple mutant. The authors directly analyzed the triple mutant without providing information on the single mir159a, mir159b, and mir159c mutants. As also suggested by point #2 below, single or double mutants might have reproductive defects that might have an influence in the triple mutant analysis. It is well known that no less than 10% (but in our hands it's more 15 to 20%) of all SALK insertional lines have reciprocal translocations that almost invariably result in female gametophytic defects (see for instance Curtis et al. 2009 *Planta* 731-745). Since crossing female mir159abc to male wild-type shows a significant fertility defect that is not further analyzed by the authors, it is impossible to discard the possibility of chromosome translocations affecting one of several of these insertional lines. Analyzing the single or double mutants with regard to nuclear proliferation in reciprocal crosses would clarify this issue before interpreting the triple mutant phenotype.

2) The phenotypic analysis of reciprocal crosses to wild-type does not distinguish gametophytic from sporophytic defects. The phenotypic analysis of reciprocal crosses does not distinguish "defective seeds" from "unfertilized ovules". Fig 1b does not allow a distinction between these two classes of defective organs within a silique. Whereas the first class refers to post-fertilization

defects, the second is indicative of defects occurring during female gametogenesis. This is particularly important for discarding the possibility that miR159 might play a role during female gametogenesis, or that the insertional lines used for generating the triple mutant contain reciprocal chromosome translocations that affect the interpretation of the results.

3) Evidence suggesting that miR159 represses maternally inherited MYB33 and MYB65 is not quantitative and should be clarified; the same is true for the subcellular localization of MYB33-YFP and mMYB33-YFP. Although results shown in Fig3 are promising, it is not clear why expression of both promoter fusions should be localized both in cytoplasm and the nucleus; can you describe which cells are you showing in the micrographs? (egg cell? central cell? nucellar cells?). A negative control and perhaps a sporophytic positive control should be presented. Finally, the authors should indicate how many ovules and seeds they observed, and how often they detect expression or not. This is also true for results presented as SupplFig4.

4) Experimental design and evidence determining the expression of MYB genes in pollen and female gametophytes is not compelling. The authors do not describe in detail how they were able to isolate and collect pollen and ovules for total RNA extraction. It is not clear at which precise developmental stages they conducted qRT-PCR experiments; were quantifications normalized?; what were the controls?. Also, the analysis of MYB33-GFP and MYB65-RFP is limited, only restricted to a couple of micrographs presented in SupplFig3c. In my opinion, the evidence showing that these two gene targets are not expressed in mature pollen, particularly sperm cells, is crucial to the conclusions of the study. Sperm cells are not visible in Suppl Fig3c. If target genes are expressed in sperm cells, the conclusion that "maternal barriers" are cleared by paternally transmitted miR159 would have to be revised.

5) Sequence of the crucial mMYB33 sequence should be provided. Since the experiments involving the transgenic mMYB33-YFP are crucial for the conclusions, I think it would be important to describe and include the sequence of the wild-type and mutant versions of the gene, as part of the Supplementary materials, indicating the precise location of the miR159 target site.

6) Why using a proFWA::DDM1-YFP reporter line for confirming abnormal nuclear divisions?. The authors do not fully interpret the experiments illustrated as the second part of Figure 2. In my opinion results provided in Fig2a to 2f are compelling. Unless something really important is missing from this manuscript with regard to a possible link between DDM1 and miR159, the inclusion of a transgenic line that over-expressed a methyltransferase under the control of a promoter from an epigenetic responding gene is really confusing, and not necessary. There could be many factors that could confound the results on the basis of well-known epigenetic phenomena occurring in developing seeds.

7) Although the discussion indicates that the authors showed mir159-dependent repression of MYB65, there is no evidence provided of a possible control of MYB65 by miR159.

8) In the discussion, the authors claim that their results "indicate that limited miRNA activity in gametes is also highly conserved from plants to animals". I don't know if evidence from a specific miRNA activity is sufficient to sustain such a broad implication.

Some additional minor comments:

9) Page5, lane 24, not clear what the authors mean by "itself as the pollen", do they mean auto-pollination of the mir159abc mutant?

10) Pgae6, lane7, replace "compare" by "compared"

11) Page9 lane 20; replace "in the ovule" by "in the developing seed"

12) Last paragraph of the discussion: it is not clear what the authors mean by "At the very beginning"; please refer to a specific developmental stage.

Finally, although the English is acceptable, I would recommend a final edition by a native English writing professional.

Reviewer #3:

Remarks to the Author:

In their manuscript Wang et al. convincingly show that microRNA miR159 delivered by sperm cells represses MYB33 and MYB65, two R2R3 MYB domain transcription factors specifically expressed in central cells. After fertilization MYB33 and MYB65 activity has to be removed allowing endosperm development. Mutated versions of MYB33 and MYB65 lacking miR159 binding sites are no longer repressed after fertilization indicated by a delay of endosperm development. Embryogenesis is not affected by absence of miR159.

Although the manuscript does not contain a large amount of data, I am very enthusiastic about the findings as they significantly further our thinking about gamete/seed activation after fertilization. However, I think the manuscript could be shortened to a Short Report, Brief Communication or similar containing two figures. Figs. 1+2 could be easily combined and Fig. 1c moved to Suppl. Mat.; Figs. 3+4 could be combined as a second figure. There is also some redundancy, which allows shortening of the manuscript: e.g. the last paragraph of the Introduction is partly a repetition of the Abstract and some introductory sentences - e.g. for the chapter "Paternal miR159..." are superfluous.

Some data should be sustained by statistics (e.g. in Fig. 1c; Fig. 3 and Fig. S4); in vitro pollen germination and growth assays are missing as well as controls for Suppl. Fig. S3c and d, which otherwise appear to display autofluorescence.

I had some problems with the English: I strongly recommend that the authors use a professional editing service before a revised version of the manuscript is resubmitted.

MINOR

-Page numbers would be helpful;

-In general they use old citations; e.g. the first general citation is from 2001 - why don't they cite more recent papers containing up-to-date knowledge?

-Fig. 1: a and b are reversed;

-Fig. 1c: they should measure seed size; legend of Fig. 1c is missing;

-Blue stars (partly also red stars) in Fig. 2a-f and Fig. 4a-f are hardly visible - I recommend using white and yellow stars; the outline of the embryo is not described in the legend of the figure;

-In Fig. 3 they should increase the contrast to improve visibility of signals;

-Suppl. Fig. S2c is very important and should be included in Fig. 2;

-They report that "only MYB33 and MYB65 were detectable" - this is wrong: these are the most strongest expressed genes in unfertilized ovules, but the other target genes were also detected;

-M&M: details about growth conditions are missing (light, humidity etc.);

-They write that in vitro pollen germination was done, but don't show data? In my opinion this should be included in Suppl. Fig. S2;

-The Discussion is very well done. I am only missing a comment/speculation why miR159 and targets are co-existing in pollen.

Responses to Referees' comments:

Reviewer #1:

This is a review of the manuscript entitled "Clearance of maternal barriers by paternal miR159 to initiate endosperm nuclear divisions in Arabidopsis" submitted to Nature Communications by Wang et al. In this manuscript, the authors demonstrate that paternal miR159 is essential to promote endosperm nuclear divisions just after fertilization. Without the paternal miR159 contribution, endosperm divisions are delayed, leading to partial sterility. The authors also identify two key targets of paternal miR159, which are maternally-expressed MYB genes. They demonstrate that miR159-resistant MYB genes provide the same endosperm phenotype as a miR159-mutant paternal contribution. Overall, the paper aims to demonstrate a rapid clearing of a maternal transcript by paternal sperm microRNAs at the moment of fertilization.

Overall, I found this manuscript very interesting, but I found the key data was often buried or very difficult to see. I have split my comments into major issues to be addressed, and smaller changes to be made.

Response: We appreciate that you find the significance of our manuscript. The presentation of our data has been improved according to your suggestions point by point in the revised manuscript. Moreover, combined with other two referees' suggestions, we carried out several additional experiments to improve the data quality of our manuscript. These experiments include 5'RACE-PCR to detect 3'cleavage products of MYB33 and MYB65 by miR159 in the developing seeds (Fig. S4), replacement of proFWA::DDM1-GFP with proFWA::RFP as the marker for endosperm (Fig. 2h-2m, S2d), quantitative analysis for seed size (Fig. 1d), and quantification analysis for almost all fluorescence images (Fig. 1c, 3d, S3d, S5b, S6b, S6c), and other minor experiments as indicated in point-by-point responses.

Major issues to be addressed

1. There are several figure panels that require quantification, rather than relying on qualitative visual inspection of the data.

A. The major weakness of this manuscript is Figure 3. I cannot tell what the authors want me to observe, and I cannot draw their same conclusions from the images. This may be a failure of communication rather than anything wrong with the data. Perhaps show the reader what is background auto-fluorescence vs what is real RFP/GFP. Or perhaps add more annotation to the figures, such as in Figure 2. Also, Quantification is needed.

Response: We are sorry for our unclear descriptions about these images, and thank you for your suggestions to provide quantification analysis. We revised figure legends to make the description more clear. As you mentioned, for representative fluorescence images, the real GFP/RFP signal is only indicated by the circle, while other distributed signals represent autofluorescence of the developing seeds. To further confirm the signal of MYB33-GFP is in the central cell, we introduced a central cell/endosperm marker line (proFWA::RFP) into proMYB33::MYB33-GFP plants to make a doubly transgenic plant, then the co-localization analysis shows that the

MYB33-GFP signal overlaps with the RFP signal (Fig. 3a). In addition, we provide quantification analysis for each category in Fig. 3d.

B. I have the same comment for Supplemental Figure 3D and Figure S4. I do not agree with the conclusions simply because I am not seeing the same thing as the authors are seeing.

Response: Sorry for our unclear labeling and description again. We rewrote figure legends for those images and provide quantification analysis for these data in Fig. S5b, S6b, S6c, respectively.

C. Figure 1C – quantification is required.

Response: Thanks. We added quantification data for this analysis in Fig. 1c.

2. I would not continue to refer to miR159 as “essential” for seed development. Some seeds are produced from a miR159 mutant x mutant cross (which is how the authors generate their key homozygous mutant line), and therefore I would write that fertility is decreased, but it is not “essential”.

Response: Thank you for your suggestion. We revised it accordingly.

3. In the discussion, please speculate on why miR159 does not act on DUO1. Why are microRNAs ineffective in the gametes, but then upon fertilization they now function? This point is raised several times in the manuscript, and why/how needs to be at least speculated on in the Discussion section.

Response: It is an interesting point. We have discussed this point in the revised version. Based on transcriptomic analysis of sperm and the female gametophytes, the possible reason is that AGO1, the core component of RISC (RNA-induced silencing complex), is very lowly expressed in gametes.

4. I find the choice of expression of DDM1 in the endosperm as a fluorescent marker very odd. Please explain why this was performed rather than using H2B or a simple NLS tag to make YFP nuclear-localized?

Response: Sorry for this confusion. There is no specific reason, just because we have proFWA::DDM1-GFP lines in our lab at that time. Based on your suggestion, we replaced it with proFWA::RFP, which is a widely used marker for the central cell and endosperm, and redid this experiment. The results are shown in Fig. 2h-2m and S2d.

5. Some of the key data should be moved from the Supplemental section to the main figures: Figure S4, Figure S2C.

Response: Thank you very much for these suggestions. To emphasize the point that disruption of paternal miR159 has no effects on activity of sperm cells, we added the data to show that no detectable role of miR159 on the release and migration of sperm from the *mir159abc* triple mutant (Fig. 1d), and remain those data in the supplementary file, including normal pollen development of the *mir159abc* triple mutant (Fig. S1b), unaffected pollen tube growth and guidance of the *mir159abc*

triple mutant (Fig. S1c), and normal female gametophyte development of the *mir159abc* triple mutant (Fig. S1d).

6. For the mMYB33-YFP transgene, details were missing in the construction: Is the mMYB33 protein identical to the MYB33 protein? Their nucleic acid sequences must be different (with and without miR159 target sites), but are their other changes that could account for their expression differences after fertilization?

Response: Sorry for our unclear descriptions. The mMYB33 protein is absolutely same to the wild type MYB33, because we mutated those nucleotides to make sure only loss of miR159 targeting capacity but without changing amino acids. We made this construct according to two papers: Millar and Gubler, The Arabidopsis GAMYB-Like genes, MYB33 and MYB65, are microRNA-regulated genes that redundantly facilitate anther development, 2005, Plant Cell, 17, 705–721; Palatnik et al., Sequence and expression differences underlie functional specialization of Arabidopsis microRNAs miR159 and miR319, 2007, Dev Cell, 13, 115–125). We added a schematic to show detailed information about this point in Fig. 4a.

7. Please perform 5' RACE to detect MYB33 & MYB65 miR159 cleavage products in the developing seeds and pollen. The detection of microRNA cleavage (and lack thereof in the mMYB33 line) would greatly strengthen the authors' arguments.

Response: It's a very good idea. Based on your suggestion, we performed 5'RACE-PCR to demonstrate that both MYB33 and MYB65 are rapidly targeted by paternal miR159 in the developing seeds after fertilization, and the results were shown in Fig. S4. Because both MYB33 and MYB65 are not expressed in mature pollen (Fig. S3a, 3c, and it was also shown in previous studies: Borges et al., Comparative transcriptomics of Arabidopsis sperm cells. 2008, Plant Physiol 148: 1168-81; Leydon et al., Three MYB transcription factors control pollen tube differentiation required for sperm release. 2013, Curr Biol 23:1209-14), we thus did not perform 5'RACE-PCR for the pollen sample.

Smaller changes to be made

1. The first sentence of the last paragraph of the Introduction needs to be rewritten with improved clarity.

Response: We revised it.

2. Supplemental Figure 1C needs to be shown with a wild-type control or reference sample. Since I do not work on pollen, I do not have a frame of reference of what a normal pistil should look like, so I cannot determine if this image is normal or not.

Response: We redid this experiment using the wild type plants (Col-0) and the *mir159abc* mutant plants together, and Fig. S1c was revised.

3. The first Results section is not critical for the rest of the manuscript, so I would suggest shortening it.

Response: Although previous studies show that reduced fertility was observed in the

mir159abc mutant and over-expression of *MYB33* or *MYB65* plants, it is not clear whether and/or to what extent of paternal, maternal, and sporophytic effects of compromised miR159 activity has a role on the fertility. We are thus supposed to survey a point that disruption of paternal miR159 activity unexpectedly caused reduced fertility, and then we go further for the regulatory mechanism.

Reviewer #2:

In flowering plants double fertilization of the egg and central cell by two independent sperm cells triggers the initiation of zygotic embryogenesis and endosperm development. Although both embryo and endosperm formation have been shown to be under the influence of many maternal factors, paternally inherited factors playing a role in seed formation have remained elusive, and no specific paternally-derived microRNA (miRNA) capable of repressing the function of maternally derived transcripts has been identified. In this manuscript Wang et al. have investigated the function of the miR159 family of miRNAs during endosperm development. They provide evidence showing that reciprocal crosses to a triple *miR159abc* insertional mutant have defects in seed formation and endosperm nuclei proliferation when *mir159abc* is used as a pollen donor. They also show that the miR159 predicted targets MYB33 and MYB65 are expressed during female but not male gametogenesis prior to fertilization, and claim that miR159 activity is responsible for abolishing their activity following fertilization. A transgenic Arabidopsis line expressing a version of MYB33 that has a mutation in the MIR159 target site shows defects in free nuclear proliferation, on the basis of which the authors claim that miR159 triggers endosperm nuclear divisions through repressing the central cell-delivered miR159 targets activates endosperm proliferation through repressing its direct targets in the central cell.

I think this is a relevant topic and study that could be of wide interest for the scientific community working on early seed development. It undoubtedly presents new findings; however, in my opinion the work is not fully convincing in its present form. Here is a list of further experimental evidence and suggested modifications that could strengthen the conclusions:

- 1) The genetic analysis of the *miR159* mutant(s) is limited to the triple mutant. The authors directly analyzed the triple mutant without providing information on the single *mir159a*, *mir159b*, and *mir159c* mutants. As also suggested by point #2 below, single or double mutants might have reproductive defects that might have an influence in the triple mutant analysis. It is well known that no less than 10% (but in our hands it's more 15 to 20%) of all SALK insertional lines have reciprocal translocations that almost invariably result in female gametophytic defects (see for instance Curtis et al. 2009 Planta 731-745). Since crossing female *mir159abc* to male wild-type shows a significant fertility defect that is not further analyzed by the authors, it is impossible to discard the possibility of chromosome translocations affecting one of several of

these insertional lines. Analyzing the single or double mutants with regard to nuclear proliferation in reciprocal crosses would clarify this issue before interpreting the triple mutant phenotype.

Response: We appreciate that you find the significance of our manuscript, and thank you very much for reminding us this important issue. You are right, many T-DNA insertional lines cause defects in female gametophyte development, but in our case, we used the wild type as the female, and pollinated by the *mir159abc* triple mutant, in which the observable defects should be theoretically only from the male side and/or post-fertilization. In addition, it is expected that the fertility of the *mir159abc* triple mutant should be much more severe than that of the *mir159ab* double mutant, if T-DNA-mediated chromosome rearrangements does significantly contribute to the defective fertility caused by compromised miR159 activity. However, both our observation and a previous finding (Allen et al., MicroR159 regulation of most conserved targets in Arabidopsis has negligible phenotypic effects, 2010, Silence 1:18) show that the *mir159abc* triple mutant is indistinguishable from the *mir159ab* double mutant from the vegetative to the reproductive stage, indicating introduction of more T-DNAs to the *mir159ab* double mutant has minor role in fertility control. We did not analyze the single mutant because expression pattern of miR159a, miR159b, and miR159c are highly overlapped, any single mutant is indistinguishable from wild type (Allen et al., Genetic analysis reveals functional redundancy and the major target genes of the Arabidopsis miR159 family, 2007, PNAS, 16371–16376; Allen et al., MicroR159 regulation of most conserved targets in Arabidopsis has negligible phenotypic effects, 2010, Silence 1:18). To further exclude the effects of T-DNA-mediated chromosome rearrangement in the *mir159abc* triple mutant on female gametophyte development, we performed DIC imaging analysis to examine the morphology of mature ovule. As shown in Fig. S1d, as the wild type ovule does, mature ovules of the *mir159abc* triple mutant harbors typical seven-cell-structure (two synergid cells, 1 egg cell, 1 central cell, and three antipodal cells). However, we could not exclude this unnoticed but important possibility that miR159 plays a role in fertility control by promoting somatic tissue development surrounding the female gametophyte, because the pistil length of the *mir159abc* triple mutant is much shorter than that of WT. We therefore cited the reference as you mentioned, and discussed related possibilities in the revised text.

2) The phenotypic analysis of reciprocal crosses to wild-type does not distinguish gametophytic from sporophytic defects. The phenotypic analysis of reciprocal crosses does not distinguish “defective seeds” from “unfertilized ovules”. Fig 1b does not allow a distinction between these two classes of defective organs within a silique. Whereas the first class refers to post-fertilization defects, the second is indicative of defects occurring during female gametogenesis. This is particularly important for discarding the possibility that miR159 might play a role during female gametogenesis, or that the insertional lines used for generating the triple mutant contain reciprocal chromosome translocations that affect the interpretation of the results.

Response: Thank you very much for these valuable suggestion. To exclude that the

reduced fertility are caused by defective female gametophyte development in the *mir159abc* triple mutant, we examined whether the mature ovule of the *mir159abc* triple mutant contains an intact seven-cell structure (the central cell, the egg cell, two synergid cells, and three antipodal cells), which is the prerequisite for seed development. Although the *mir159abc* triple mutant has short pistils with only ~30 developing seeds per pistil (40-50 developing seeds per silique in wild type), we showed that a clearly visible seven-cell structure of mature ovules from the *mir159abc* triple mutant (Fig. S1d). Moreover, the capacity of the *mir159abc* triple mutant attracting pollen tubes is indistinguishable from that of WT (Fig. S1c), and the release and migration of twin sperm of the *mir159abc* triple mutant is quite normal (Fig. 1d). These results indicate that compromised miR159 activity has minor role on female gametophyte development. However, because loss of maternal miR159 also shows a significant fertility defect that is not further analyzed by us, it is worthy to investigate how maternal miR159 regulates seed development in the future.

3) Evidence suggesting that miR159 represses maternally inherited MYB33 and MYB65 is not quantitative and should be clarified; the same is true for the subcellular localization of MYB33-YFP and mMYB33-YFP. Although results shown in Fig3 are promising, it is not clear why expression of both promoter fusions should be localized both in cytoplasm and the nucleus; can you describe which cells are you showing in the micrographs? (egg cell? central cell? nucellar cells?). A negative control and perhaps a sporophytic positive control should be presented. Finally, the authors should indicate how many ovules and seeds they observed, and how often they detect expression or not. This is also true for results presented as SupplFig4.

Response: We are sorry for our unclear descriptions about these fluorescence images, and thank you for your suggestion to provide quantification analysis. We revised figure legends to make the description more clear, in which we emphasized that the real GFP/RFP signal is only indicated by the circle, while other distributed signal represents autofluorescence of ovule. To further confirm the signal of MYB33-GFP is in the central cell, we introduced a central cell/endosperm marker line (proFWA::RFP) into proMYB33::MYB33-GFP plants to make a doubly transgenic plants, then the co-localization analysis demonstrates that the GFP signal overlaps with the RFP signal (Fig. 3a). In addition, we provide quantification analysis for each category in the revised version (Fig. 3d, S3d), and added information in figures about how many (n) developing seeds we observed, and how often (%) we detected expression of MYB33 and MYB65.

4) Experimental design and evidence determining the expression of MYB genes in pollen and female gametophytes is not compelling. The authors do not describe in detail how they were able to isolate and collect pollen and ovules for total RNA extraction. It is not clear at which precise developmental stages they conducted qRT-PCR experiments; were quantifications normalized?; what were the controls?. Also, the analysis of MYB33-GFP and MYB65-RFP is limited, only restricted to a couple of micrographs presented in SupplFig3c. In my opinion, the evidence showing

that these two gene targets are not expressed in mature pollen, particularly sperm cells, is crucial to the conclusions of the study. Sperm cells are not visible in Suppl Fig3c. If target genes are expressed in sperm cells, the conclusion that “maternal barriers” are cleared by paternally transmitted miR159 would have to be revised.

Response: We have added detailed information about pollen and ovule collection in the methods. Figure S3c showed that MYB33 and MYB65 were detected in neither pollen nor sperm, those evenly distributed signal represents autofluorescence of mature pollen. Moreover, the finding of the absence of MYB33 and MYB65 in pollen, particularly in sperm, is also shown by two previous studies (Borges et al., Comparative transcriptomics of Arabidopsis sperm cells. 2008, *Plant Physiol* 148: 1168-81; Leydon et al., Three MYB transcription factors control pollen tube differentiation required for sperm release. 2013, *Curr Biol* 23:1209-14).

5) Sequence of the crucial mMYB33 sequence should be provided. Since the experiments involving the transgenic mMYB33-YFP are crucial for the conclusions, I think it would be important to describe and include the sequence of the wild-type and mutant versions of the gene, as part of the Supplementary materials, indicating the precise location of the miR159 target site.

Response: Sorry for our unclear descriptions. The mMYB33 protein is absolutely same to the wild type MYB33 one, because we only mutated the nucleotides to make sure only the loss of miR159 binding capacity but without changing any amino acids. We provided a schematic showing detailed information about this point in Fig. 4a.

6) Why using a proFWA::DDM1-YFP reporter line for confirming abnormal nuclear divisions?. The authors do not fully interpret the experiments illustrated as the second part of Figure 2. In my opinion results provided in Fig2a to 2f are compelling. Unless something really important is missing from this manuscript with regard to a possible link between DDM1 and miR159, the inclusion of a transgenic line that over-expressed a methyltransferase under the control of a promoter from an epigenetic responding gene is really confusing, and not necessary. There could be many factors that could confound the results on the basis of well-known epigenetic phenomena occurring in developing seeds.

Response: Sorry for this confusion. The same concern was also pointed by Reviewer 1. There is no specific reason, just because we have proFWA::DDM1-GFP lines in our lab at that time. There is no any specific connection between DDM1 and miR159, we only want to use an endosperm marker to indicate the failure and the delay of endosperm nuclear divisions in the loss of paternal miR159. According to your suggestion, we thus replaced it with proFWA::RFP marker line, which is a widely used maker for the central cell and endosperm, and redid this experiment. The results were shown in Fig. 2h-2m, S2d.

7) Although the discussion indicates that the authors showed mir159-dependent repression of MYB65, there is no evidence provided of a possible control of MYB65 by miR159.

Response: We show that the RFP signal of proMYB65::MYB65-RFP plants is controlled by paternal miR159, because disruption of paternal miR159 causes retention of MYB65-RFP in the central cell after fertilization (Fig. S3d). In addition, in the revised manuscript, we performed 5'RACE-PCR experiments to demonstrate the cleavage of maternal MYB65 by paternal miR159 occurs quickly in the developing seeds after fertilization (Fig. S4).

8) In the discussion, the authors claim that their results “indicate that limited miRNA activity in gametes is also highly conserved from plants to animals”. I don't know if evidence from a specific miRNA activity is sufficient to sustain such a broad implication.

Response: It's an interesting point. We revised the text according to your suggestion. Actually, there are also evidences showing limited miRNA activities in mouse oocytes (Suh, N. *et al.* MicroRNA function is globally suppressed in mouse oocytes and early embryos. *Curr Biol* 20, 271-277 (2010); Ma, J. *et al.* MicroRNA activity is suppressed in mouse oocytes. *Curr Biol* 20, 265-270 (2010)). We discussed that the potential reason is that AGO1, a core component for miRNA activity, is very lowly expressed in gametes.

Some additional minor comments:

9) Page5, line 24, not clear what the authors mean by “itself as the pollen”, do they mean auto-pollination of the mir159abc mutant?

Response: We corrected it.

10) Page6, line7, replace “compare” by “compared”

Response: We corrected it.

11) Page9 line 20; replace “in the ovule” by “in the developing seed”

Response: We corrected it.

12) Last paragraph of the discussion: it is not clear what the authors mean by “At the very beginning”; please refer to a specific developmental stage.

Response: We revised it to “within 10 hours after fertilization”.

Finally, although the English is acceptable, I would recommend a final edition by a native English writing professional.

Response: We asked Prof. Sheila McCormick (a scientist in pollen biology, UC-Berkeley) to edit our manuscript to improve the English.

Reviewer #3 (Remarks to the Author):

In their manuscript Wang et al. convincingly show that microRNA miR159 delivered by sperm cells represses MYB33 and MYB65, two R2R3 MYB domain transcription factors specifically expressed in central cells. After fertilization MYB33 and MYB65

activity has to be removed allowing endosperm development. Mutated versions of MYB33 and MYB65 lacking miR159 binding sites are no longer repressed after fertilization indicated by a delay of endosperm development. Embryogenesis is not affected by absence of miR159.

Although the manuscript does not contain a large amount of data, I am very enthusiastic about the findings as they significantly further our thinking about gamete/seed activation after fertilization. However, I think the manuscript could be shortened to a Short Report, Brief Communication or similar containing two figures. Figs. 1+2 could be easily combined and Fig. 1c moved to Suppl. Mat.; Figs. 3+4 could be combined as a second figure. There is also some redundancy, which allows shortening of the manuscript: e.g. the last paragraph of the Introduction is partly a repetition of the Abstract and some introductory sentences - e.g. for the chapter "Paternal miR159..." are superfluous.

Response: We appreciate that you find the significance of our manuscript. Based on your suggestion, we seriously discussed with the editor whether we should shorten our manuscript or not. Considering that our figures contain a lot of visual information, combining them would further reduce the size of each element and potentially make the data harder to see, so it might be better to keep the current length of the manuscript.

Some data should be sustained by statistics (e.g. in Fig. 1c; Fig. 3 and Fig. S4); in vitro pollen germination and growth assays are missing as well as controls for Suppl. Fig. S3c and d, which otherwise appear to display autofluorescence.

Response: Thank you very much for your suggestion. We provide quantification data for all these analysis in the revised version (Figures 1c, 3d, S3d, S5b, S6b, S6c). We redid in vivo pollen tube growth and guidance experiment using the wild type (Col-0) and the *mir159abc* mutant plants, the wild-type control image has been added in Fig. S1c. Besides providing statistic analysis, we revised figure legends for Fig. S3c and S5a to make it more clear.

I had some problems with the English: I strongly recommend that the authors use a professional editing service before a revised version of the manuscript is resubmitted.

Response: We asked Prof. Sheila McCormick (a scientist in pollen biology, UC-Berkeley) to edit our manuscript to improve the English.

MINOR

-Page numbers would be helpful;

Response: We did it in the revised version.

-In general they use old citations; e.g. the first general citation is from 2001 - why don't they cite more recent papers containing up-to-date knowledge?

Response: We corrected it in the revised version.

-Fig. 1: a and b are reversed;

Response: We corrected it in the revised version.

-Fig. 1c: they should measure seed size; legend of Fig. 1c is missing;

Response: Sorry for these mistakes. We provided statistic analysis for seed size (Fig. 1c), and we added figure legends of Fig. 1c in the revised version.

-Blue stars (partly also red stars) in Fig. 2a-f and Fig. 4a-f are hardly visible - I recommend using white and yellow stars; the outline of the embryo is not described in the legend of the figure;

Response: Thank you for your suggestions. We revised these information as you suggested.

-In Fig. 3 they should increase the contrast to improve visibility of signals;

Response: Thank you for your suggestion. We provided a central cell/endosperm marker (proFWA::RFP) to show the expression of MYB33 in the central cell (Fig. 3a), and provide quantitative analysis for each category of MYB33-GFP signal intensity (Fig. 3d).

-Suppl. Fig. S2c is very important and should be included in Fig. 2;

Response: Thanks for your suggestion. We revised it.

-They report that "only MYB33 and MYB65 were detectable" - this is wrong: these are the most strongest expressed genes in unfertilized ovules, but the other target genes were also detected;

Response: Thanks for your reminder. We corrected it.

-M&M: details about growth conditions are missing (light, humidity etc.);

Response: We added it.

-They write that in vitro pollen germination was done, but don't show data? In my opinion this should be included in Suppl. Fig. S2;

Response: We were supposed to show both in vitro and in vivo pollen tube growth analysis, but later we think it is enough to show in vivo data. Sorry for this mistake, and we revised it.

-The Discussion is very well done. I am only missing a comment/speculation why miR159 and targets are co-existing in pollen.

Response: It is an interesting point. Both Reviewer 1 and 2 also mentioned it. We discussed this point in the revised text. Based on transcriptomic analysis of sperm and the female gametophytes, the possible reason is that AGO1, the core component of RISC (RNA-induced silencing complex), is very lowly expressed in gametes.

Reviewers' Comments:

Reviewer #1:

Remarks to the Author:

This is a re-review of the manuscript entitled "Clearance of maternal barriers by paternal miR159 to initiate endosperm nuclear division in Arabidopsis" by Dr Zheng and colleagues. The revised manuscript is greatly improved and has made significant progress. However, the authors have failed to perform a critical non-transgenic control that I requested in my original review. See below.

1. In Figure 3, The authors have not added a wt non-transgenic sample as I requested. The background fluorescence is very high in the images presented, and this non-transgenic control is necessary for the reader to determine what is auto-fluorescence and what is real GFP/RFP signal. The authors may understand the distinction, but this cannot be taken as an assumption, and the signal to noise determination must be made scientifically, rather than by trust. I suggest imaging non-transgenic plants, then overlaying the images: the signal from the background will be very easily distinguished from the real GFP/RFP fluorescence if this approach is performed. This same comment holds true for Fig S5 as well.
2. In Fig 3D, I could not tell if the "O HAP Col-0" sample has a transgene, or is a fully wt plant? The labelling of the exact genotypes, rather than the inferred genotype, should be made throughout the manuscript.
3. Fig S4 is very important for identifying the direct nature of the regulation of the MYB genes by miR159. This supplemental figure should be moved to the main text.
4. The data accessibility policy of this paper is not acceptable. The authors indicate that "any reasonable request" for data will be provided, however, who will judge "reasonability"? The authors? That is a very large conflict of interest. The data release policy should state that all requests for data will be met.

Reviewer #2:

Remarks to the Author:

I have concentrated in revising how the authors addressed the issues that I previously raised. Whereas many of them are in my opinion addressed to satisfaction for future expert readers, there are three issues for which I still don't find sufficient clarification:

- a) There is no evidence demonstrating that some of the double mutant combinations do or do not have an equivalent phenotype to the triple miR159abc mutant. This is important not only to discard possible effects from chromosome rearrangements (as mentioned in their response), but also to provide a framework to other researchers interested in repeating or using findings from this interesting study. A double mutant combination showing the phenotype would discard the necessity to work with a triple mutant. Providing cytological or genetic evidence for at least one of the double mutant combinations should be sufficient.
- 2) The authors should clearly state in the discussion if they believe the triple (or double by all means) mutant combination shows a defect in female reproductive development. This is essential for future work willing to further investigate an exclusive paternal, or maternal and maternal role of the miR159 family.
- 3) Finally, I still don't see sufficient description on how the ovules or developing seeds were collected at stages that are not clearly defined as part of the results presented in Supplementary Figure4 (qRT-PCR). Again, how were the ovules and developing seeds (either before or after pollination) isolated and collected? I assume HAP is hours after pollination, but it is not explicitly mentioned. How were the gynoecea pollinated? which genotypes were the parents?

Reviewer #3:

Remarks to the Author:

The authors have made a very serious attempt to improve their manuscript. With the exception of the partly outdated references all my comments have been addressed satisfactorily. The manuscript has improved a lot and the added statistics are solid. In case there is a final revision the authors may consider to further refresh the literature.

Reviewer #1 (Remarks to the Author):

This is a re-review of the manuscript entitled "Clearance of maternal barriers by paternal miR159 to initiate endosperm nuclear division in Arabidopsis" by Dr Zheng and colleagues. The revised manuscript is greatly improved and has made significant progress. However, the authors have failed to perform a critical non-transgenic control that I requested in my original review. See below.

Response: Thank you very much for your consideration and suggestion. We have added the critical auto-fluorescence control or the non-transgenic control for each fluorescence image, and revised the figure legends and text accordingly point by point in the second-round revised manuscript.

1. In Figure 3, The authors have not added a wt non-transgenic sample as I requested. The background fluorescence is very high in the images presented, and this non-transgenic control is necessary for the reader to determine what is auto-fluorescence and what is real GFP/RFP signal. The authors may understand the distinction, but this cannot be taken as an assumption, and the signal to noise determination must be made scientifically, rather than by trust. I suggest imaging non-transgenic plants, then overlaying the images: the signal from the background will be very easily distinguished from the real GFP/RFP fluorescence if this approach is performed. This same comment holds true for Fig S5 as well.

Response: Sorry for our mistakes to ignore this important issue. We have provided the control auto-fluorescence image to help readers distinguish the real signal from the background fluorescence for each category in the revised manuscript.

2. In Fig 3D, I could not tell if the "0 HAP Col-0" sample has a transgene, or is a fully wt plant? The labelling of the exact genotypes, rather than the inferred genotype, should be made throughout the manuscript.

Response: Sorry for our unclear description. "0 HAP Col-0" indicates the wt carrying the MYB33-GFP transgene, and "0 HAP" means unfertilized ovule/mature ovule. We have revised the labels in the revised manuscript.

3. Fig S4 is very important for identifying the direct nature of the regulation of the MYB genes by miR159. This supplemental figure should be moved to the main text.

Response: Thank you for your suggestion. We moved it to the main text as Fig 4.

4. The data accessibility policy of this paper is not acceptable. The authors indicate that "any reasonable request" for data will be provided, however, who will judge "reasonability"? The authors? That is a very large conflict of interest. The data release policy should state that all requests for data will be met.

Response: Sorry for our incorrect description. This study has no any conflict of interest, and it is definitely sure that all requests for data will be met. We revised it accordingly.

Reviewer #2 (Remarks to the Author):

I have concentrated in revising how the authors addressed the issues that I previously raised. Whereas many of them are in my opinion addressed to satisfaction for future expert readers, there are three issues for which I still don't find sufficient clarification:

a) There is no evidence demonstrating that some of the double mutant combinations do or do not have an equivalent phenotype to the triple *miR159abc* mutant. This is important not only to discard possible effects from chromosome rearrangements (as mentioned in their response), but also to provide a framework to other researchers interested in repeating or using findings from this interesting study. A double mutant combination showing the phenotype would discard the necessity to work with a triple mutant. Providing cytological or genetic evidence for at least one of the double mutant combinations should be sufficient.

Response: Thank you again for this suggestion. We isolated the double mutant of *mir159ab*, *mir159ac*, and *mir159bc* from the progenies of the *mir159abc* triple mutant backcrossing with the wild type Col-0 plants, respectively. Among three double mutants, the *mir159ab* plants exhibit indistinguishable morphology with that of the *mir159abc* triple mutant (Supplementary Fig. 2d), while other two double mutants (*mir159ac* and *mir159bc*) and all single mutants are indistinguishable from wild type plants (Supplementary Fig. 2d), indicating that miR159a and miR159b play a dominant and redundant role in plant development. We then performed DIC analysis to investigate whether endosperm nuclear divisions in the *mir159ab* double mutant are affected as does in the *mir159abc* triple mutant. Indeed, we observed that when the *mir159ab* double mutant was used as the pollen to pollinate Col-0 pistil, endosperm nuclear divisions of progenies were clearly arrested and/or delayed (Supplementary Fig. 2e), and the severity of defective endosperm nuclear division was comparable between the *mir159ab* double mutant and the *mir159abc* triple mutant (Fig. 2g and Supplementary Fig. 2e).

2) The authors should clearly state in the discussion of they believe the triple (or double by all means) mutant combination shows a defect in female reproductive development. This is essential for future work willing to further investigate an exclusive paternal, or maternal and maternal role of the miR159 family.

Response: We added the related discussion about the potential role of miR159 in female gametophyte development. We indeed observed similar defects in the transmission of female gametes (Supplementary Table 1), indicating that maternal miR159 is required for the transmission of female gametes. Moreover, based on the observation of short pistils in the *mir159ab* double and the *mir159abc* triple mutants, we surmise that maternal miR159 is involved in female reproductive development, but the mechanism needs further investigations. Thank you for your suggestion.

3) Finally, I still don't see sufficient description on how the ovules or developing

seeds were collected at stages that are not clearly defined as part of the results presented in Supplementary Figure 4 (qRT-PCR). Again, how were the ovules and developing seeds (either before or after pollination) isolated and collected? I assume HAP is hours after pollination, but it is not explicitly mentioned. How were the gynoecea pollinated? which genotypes were the parents?

Response: Sorry for our unclear descriptions. We picked the developing seeds from dissected pistils before or after fertilization as indicated time points, and put the developing seeds in the liquid nitrogen immediately. We provided the abbreviation for HAP in the text. For all pollination manipulations, we usually let the naked pistil by removing all organs (sepals, petals, and stamens) for ~12 hours, and then pollinate the naked pistil using mature pollen from indicated genotypes. For Supplementary Fig. 4c, we used pistils from the wild type plants Col-0, and pollinated it using mature pollen from Col-0 or the *mir159abc* triple mutant. For Supplementary Fig. 4d, we used pistils from the proFWA::mMYB33-GFP transgenic plants, and pollinated it using mature pollen from Col-0 plants.

Reviewer #3 (Remarks to the Author):

The authors have made a very serious attempt to improve their manuscript. With the exception of the partly outdated references all my comments have been addressed satisfactorily. The manuscript has improved a lot and the added statistics are solid. In case there is a final revision the authors may consider to further refresh the literature.

Response: Thank you very much! We updated references from recent progress in this field.

Reviewers' Comments:

Reviewer #1:

Remarks to the Author:

The authors have fulfilled all of my remaining comments.

Reviewer #2:

None